# Reward is Enough for Convex MDPs

**Tom Zahavy**
DeepMind, London
tomzahavy@deepmind.com

**Brendan O'Donoghue**
DeepMind, London
bodonoghue@deepmind.com

**Guillaume Desjardins**
DeepMind, London
gdesjardins@deepmind.com

**Satinder Singh**
DeepMind, London
baveja@deepmind.com

## Abstract

Maximising a cumulative reward function that is Markov and stationary, *i.e.*, defined over state-action pairs and independent of time, is sufficient to capture many kinds of goals in a Markov decision process (MDP). However, not all goals can be captured in this manner. In this paper we study convex MDPs in which goals are expressed as convex functions of the stationary distribution and show that they cannot be formulated using stationary reward functions. Convex MDPs generalize the standard reinforcement learning (RL) problem formulation to a larger framework that includes many supervised and unsupervised RL problems, such as apprenticeship learning, constrained MDPs, and so-called 'pure exploration'. Our approach is to reformulate the convex MDP problem as a min-max game involving policy and cost (negative reward) 'players', using Fenchel duality. We propose a meta-algorithm for solving this problem and show that it unifies many existing algorithms in the literature.

## 1 Introduction

In reinforcement learning (RL), an agent learns how to map situations to actions so as to maximize a cumulative scalar reward signal. The learner is not told which actions to take, but instead must discover which actions lead to the most reward [64]. Mathematically, the RL problem can be written as finding a policy whose state occupancy has the largest inner product with a reward vector [55], *i.e.*, the goal of the agent is to solve

$$\text{RL:} \quad \max_{d_\pi \in \mathcal{K}} \sum_{s,a} r(s,a) d_\pi(s,a), \tag{1}$$

where $d_\pi$ is the state-action stationary distribution induced by policy $\pi$ and $\mathcal{K}$ is the set of admissible stationary distributions (see Definition 1). A significant body of work is dedicated to solving the RL problem efficiently in challenging domains [45, 62]. However, not all decision making problems of interest take this form. In particular we consider the more general *convex* MDP problem,

$$\text{Convex MDP:} \quad \min_{d_\pi \in \mathcal{K}} f(d_\pi), \tag{2}$$

where $f : \mathcal{K} \to \mathbb{R}$ is a convex function. Sequential decision making problems that take this form include Apprenticeship Learning (AL), pure exploration, and constrained MDPs, among others; see Table 1. In this paper we prove the following claim:

*We can solve Eq.* (2) *by using any algorithm that solves Eq.* (1) *as a subroutine.*

In other words, any algorithm that solves the standard RL problem can be used to solve the more general convex MDP problem. More specifically, we make the following contributions.

35th Conference on Neural Information Processing Systems (NeurIPS 2021).

**Firstly**, we adapt the meta-algorithm of Abernethy and Wang [3] for solving Eq. (2). The key idea is to use Fenchel duality to convert the convex MDP problem into a two-player zero-sum game between the agent (henceforth, *policy player*) and an adversary that produces rewards (henceforth, *cost player*) that the agent must maximize [3, 6]. From the agent's point of view, the game is bilinear, and so for fixed rewards produced by the adversary the problem reduces to the standard RL problem with non-stationary reward (Fig. 1).

**Secondly**, we propose a sample efficient policy player that uses a standard RL algorithm (*eg*, [35, 60]), and computes an optimistic policy with respect to the non-stationary reward at each iteration. In other words, we use algorithms that were developed to achieve low regret in the standard RL setup, to achieve low regret as policy players in the min-max game we formulate to solve the convex MDP. Our main result is that the average of the policies produced by the policy player converges to a solution to the convex MDP problem (Eq. (2)). Inspired by this principle, we also propose a recipe for using deep-RL

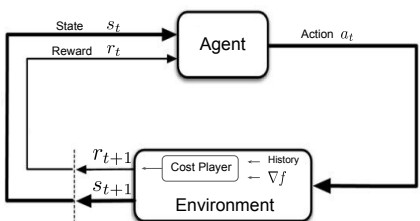

Figure 1: Convex MDP as an RL problem

(DRL) agents to solve convex MDPs heuristically: provide the agent non-stationary rewards from the cost player. We explore this principle in our experiments.

**Finally**, we show that choosing specific algorithms for the policy and cost players unifies several disparate branches of RL problems, such as apprenticeship learning, constrained MDPs, and pure exploration into a single framework, as we summarize in Table 1.

| Convex objective $f$ | Cost player | Policy player | Application |
|---|---|---|---|
| $\lambda \cdot d_\pi$ | FTL | RL | (Standard) RL with $-\lambda$ as stationary reward function |
| $\|d_\pi - d_E\|_2^2$ | FTL | Best response | Apprenticeship learning (AL) [1, 75] |
| $d_\pi \cdot \log(d_\pi)$ | FTL | Best response | Pure exploration* [30] |
| $\|d_\pi - d_E\|_\infty$ | OMD | Best response | AL [66, 65] |
| $\mathbb{E}_c\left[\lambda c \cdot (d_\pi - d_E(c))\right]^\dagger$ | OMD | Best response | Inverse RL in contextual MDPs [10] |
| $\lambda_1 \cdot d_\pi, \text{ s.t. } \lambda_2 \cdot d_\pi \leq c$ | OMD | RL | Constrained MDPs [7, 67, 12, 68, 18, 16, 11] |
| $\text{dist}(d_\pi, C)^{\dagger\dagger}$ | OMD | Best response | Feasibility of convex-constrained MDPs [44] |
| $\min_{\lambda_1,\ldots,\lambda_k} d_\pi^k \cdot \lambda_k$ | OMD | RL | Adversarial Markov Decision Processes [57] |
| $\max_{\lambda \in \Lambda} \lambda \cdot (d_\pi - d_E)$ | OMD | RL | Online AL [61], Wasserstein GAIL [73, 78] |
| $\text{KL}(d_\pi \| d_E)$ | FTL | RL | GAIL [31], state marginal matching [41], |
| $-\mathbb{E}_z \text{KL}(d_\pi^z \| \mathbb{E}_k d_\pi^k)^\ddagger$ | FTL | RL | Diverse skill discovery [26, 20, 27, 21, 69, 4] |

Table 1: Instances of Algorithm 1 in various convex MDPs. * as well as other KL divergences. $^\dagger$ $c$ is a context variable. $^{\dagger\dagger}$ $C$ is a convex set. $^\ddagger$ $f$ is concave. See Sections 4 & 6 for more details.

## 2 Reinforcement Learning Preliminaries

In RL an agent interacts with an environment over a number of time steps and seeks to maximize its cumulative reward. We consider two cases, the average reward case and the discounted case. The Markov decision process (MDP) is defined by the tuple $(S, A, P, R)$ for the average reward case and by the tuple $(S, A, P, R, \gamma, d_0)$ for the discounted case. We assume an infinite horizon, finite state-action problem where initially, the state of the agent is sampled according to $s_0 \sim d_0$, then at each time $t$ the agent is in state $s_t \in S$, selects action $a_t \in A$ according to some policy $\pi(s_t, \cdot)$, receives reward $r_t \sim R(s_t, a_t)$ and transitions to new state $s_{t+1} \in S$ according to the probability distribution $P(\cdot, s_t, a_t)$. The two performance metrics we consider are given by

$$J_\pi^{\text{avg}} = \lim_{T \to \infty} \frac{1}{T} \mathbb{E} \sum_{t=1}^T r_t, \quad J_\pi^\gamma = (1-\gamma)\mathbb{E} \sum_{t=1}^\infty \gamma^t r_t, \tag{3}$$

for the average reward case and discounted case respectively. The goal of the agent is to find a policy that maximizes $J_\pi^{\text{avg}}$ or $J_\pi^\gamma$. Any stationary policy $\pi$ induces a *state-action occupancy measure* $d_\pi$,

which measures how often the agent visits each state-action when following $\pi$. Let $\mathbb{P}_\pi(s_t = \cdot)$ be the probability measure over states at time $t$ under policy $\pi$, then

$$d_\pi^{\mathrm{avg}}(s, a) = \lim_{T \to \infty} \frac{1}{T} \mathbb{E} \sum_{t=1}^{T} \mathbb{P}_\pi(s_t = s)\pi(s, a), \ \ d_\pi^\gamma(s, a) = (1 - \gamma)\mathbb{E} \sum_{t=1}^{\infty} \gamma^t \mathbb{P}_\pi(s_t = s)\pi(s, a),$$

for the average reward case and the discounted case respectively. With these, we can rewrite the RL objective in Eq. (3) in terms of the occupancy measure using the following well-known result, which for completeness we prove in Appendix B.

**Proposition 1.** *For both the average and the discounted case, the agent objective function Eq. (3) can be written in terms of the occupancy measure as $J_\pi = \sum_{s,a} r(s, a)d_\pi(s, a)$.*

Given an occupancy measure it is possible to recover the policy by setting $\pi(s, a) = d_\pi(s, a)/\sum_a d_\pi(s, a)$ if $\sum_a d_\pi(s, a) > 0$, and $\pi(s, a) = 1/|A|$ otherwise. Accordingly, in this paper we shall formulate the RL problem using the state-action occupancy measure, and both the standard RL problem (Eq. (1)) and the convex MDP problem (Eq. (2)) are convex optimization problems in variable $d_\pi$. For the purposes of this manuscript we do not make a distinction between the average and discounted settings, other than through the convex polytopes of feasible occupancy measures, which we define next.

**Definition 1** (State-action occupancy's polytope [55])**.** *For the average reward case the set of admissible state-action occupancies is*

$$\mathcal{K}_{\mathrm{avg}} = \{d_\pi \mid d_\pi \geq 0, \ \sum_{s,a} d_\pi(s, a) = 1, \ \sum_a d_\pi(s, a) = \sum_{s',a'} P(s, s', a')d_\pi(s', a') \ \ \forall s \in S\},$$

*and for the discounted case it is given by*

$$\mathcal{K}_\gamma = \{d_\pi \mid d_\pi \geq 0, \ \sum_a d_\pi(s, a) = (1 - \gamma)d_0(s) + \gamma \sum_{s',a'} P(s, s', a')d_\pi(s', a') \ \ \forall s \in S\}.$$

We note that being a polytope implies that $\mathcal{K}$ is a convex and compact set.

The convex MDP problem is defined for the tuple $(S, A, P, f)$ in the average cost case and $(S, A, P, f, \gamma, d_0)$ in the discounted case. This tuple is defining a state-action occupancy's polytope $\mathcal{K}$ (Definition 1), and the problem is to find a policy $\pi$ whose state occupancy $d_\pi$ is in this polytope and minimizes the function $f$ (Eq. (2)).

## 3  A Meta-Algorithm for Solving Convex MDPs via RL

To solve the convex MDP problem (Eq. (2)) we need to find an occupancy measure $d_\pi$ (and associated policy) that minimizes the function $f$. Since both $f : \mathcal{K} \to \mathbb{R}$ and the set $\mathcal{K}$ are convex this is a convex optimization problem. However, it is a challenging one due to the nature of learning about the environment through stochastic interactions. In this section we show how to reformulate the convex MDP problem (Eq. (2)) so that standard RL algorithms can be used to solve it, allowing us to harness decades of work on solving vanilla RL problems. To do that we will need the following definition.

**Definition 2** (Fenchel conjugate)**.** *For a function $f : \mathbb{R}^n \to \mathbb{R} \cup \{-\infty, \infty\}$, its Fenchel conjugate is denoted $f^* : \mathbb{R}^n \to \mathbb{R} \cup \{-\infty, \infty\}$ and defined as $f^*(x) := \sup_y x \cdot y - f(y)$.*

**Remark 1.** *The Fenchel conjugate function $f^*$ is always convex (when it exists) even if $f$ is not. Furthermore, the biconjugate $f^{**} := (f^*)^*$ equals $f$ if and only if $f$ is convex and lower semi-continuous.*

Using this we can rewrite the convex MDP problem (Eq. (2)) as

$$f^{\mathrm{OPT}} = \min_{d_\pi \in \mathcal{K}} f(d_\pi) = \min_{d_\pi \in \mathcal{K}} \max_{\lambda \in \Lambda} (\lambda \cdot d_\pi - f^*(\lambda)) = \max_{\lambda \in \Lambda} \min_{d_\pi \in \mathcal{K}} (\lambda \cdot d_\pi - f^*(\lambda)) \tag{4}$$

where $\Lambda$ is the closure of (sub-)gradient space $\{\partial f(d_\pi)|d_\pi \in \mathcal{K}\}$, which is a convex set [3, Theorem 4]. As both sets are convex, this is a convex-concave saddle-point problem and a zero-sum two-player game [54, 49], and we were able to swap the order of minimization and maximization using the minimax theorem [71].

With this we define the Lagrangian as $\mathcal{L}(d_\pi, \lambda) := \lambda \cdot d_\pi - f^*(\lambda)$. For a fixed $\lambda \in \Lambda$, minimizing the Lagrangian is a standard RL problem of the form of Eq. (1), *i.e.*, equivalent to maximizing a reward $r = -\lambda$. Thus, one might hope that by producing an optimal dual variable $\lambda^\star$ we could simply solve $d_\pi^\star = \arg\min_{d_\pi \in \mathcal{K}} \mathcal{L}(\cdot, \lambda^\star)$ for the optimal occupancy measure. However, the next lemma states that this is not possible in general.

**Lemma 1.** *There exists an MDP $M$ and convex function $f$ for which there is no stationary reward $r \in \mathbb{R}^{S \times A}$ such that $\arg\max_{d_\pi \in \mathcal{K}} d_\pi \cdot r = \arg\min_{d_\pi \in \mathcal{K}} f(d_\pi)$.*

To see this note that for any reward $r$ there is a deterministic policy that optimizes the reward [55], but for some choices of $f$ no deterministic policy is optimal, *eg*, when $f$ is the negative entropy function. This result tells us that even if we have access to an optimal dual-variable we cannot simply use it to recover the stationary distribution that solves the convex MDP problem in general.

To overcome this issue we develop an algorithm that generates a *sequence* of policies $\{\pi^k\}_{k \in \mathbb{N}}$ such that the average converges to an optimal policy for Eq. (2), *i.e.*, $(1/K) \sum_{k=1}^{K} d_\pi^k \to d_\pi^\star \in \arg\min_{d_\pi \in \mathcal{K}} f(d_\pi)$. The algorithm we develop is described in Algorithm 1 and is adapted from the meta-algorithm described in Abernethy and Wang [3]. It is referred to as a *meta-algorithm* since it relies on supplied sub-routine algorithms $\text{Alg}_\pi$ and $\text{Alg}_\lambda$. The reinforcement learning algorithm $\text{Alg}_\pi$ takes as input a reward vector and returns a state-action occupancy measure $d_\pi$. The cost algorithm $\text{Alg}_\lambda$ can be a more general function of the entire history. We discuss concrete examples of $\text{Alg}_\pi$ and $\text{Alg}_\lambda$ in Section 4.

---

**Algorithm 1:** meta-algorithm for convex MDPs

1: **Input:** convex-concave payoff $\mathcal{L} : \mathcal{K} \times \Lambda \to \mathcal{R}$, algorithms $\text{Alg}_\lambda, \text{Alg}_\pi, K \in \mathbb{N}$
2: **for** $k = 1, \dots, K$ **do**
3:    $\lambda^k = \text{Alg}_\lambda(d_\pi^1, \dots, d_\pi^{k-1}; \mathcal{L})$
4:    $d_\pi^k = \text{Alg}_\pi(-\lambda^k)$
5: **end for**
6: Return $\bar{d}_\pi^K = \frac{1}{K} \sum_{k=1}^{K} d_\pi^k, \bar{\lambda}^K = \frac{1}{K} \sum_{k=1}^{K} \lambda^k$

---

In order to analyze this algorithm we will need a small detour into online convex optimization (OCO). In OCO, a learner is presented with a sequence of $K$ convex loss functions $\ell_1, \ell_2, \dots, \ell_K : \mathcal{K} \to \mathbb{R}$ and at each round $k$ must select a point $x_k \in \mathcal{K}$ after which it suffers a loss of $\ell_k(x_k)$. At time period $k$ the learner is assumed to have perfect knowledge of the loss functions $\ell_1, \dots, \ell_{k-1}$. The learner wants to minimize its *average regret*, defined as

$$\bar{R}_K := \frac{1}{K} \left( \sum_{k=1}^{K} \ell_k(x_k) - \min_{x \in \mathcal{K}} \sum_{k=1}^{K} \ell_k(x) \right).$$

In the context of convex reinforcement learning and meta-algorithm 1, the loss functions for the cost player are $\ell_\lambda^k = -\mathcal{L}(\cdot, \lambda^k)$, and for the policy player are $\ell_\pi^k = \mathcal{L}(d_\pi^k, \cdot)$, with associated average regrets $\bar{R}_K^\pi$ and $\bar{R}_K^\lambda$. This brings us to the following theorem.

**Theorem 1** (Theorem 2, [3]). *Assume that $\text{Alg}_\pi$ and $\text{Alg}_\lambda$ have guaranteed average regret bounded as $\bar{R}_K^\pi \le \epsilon_K$ and $\bar{R}_K^\lambda \le \delta_K$, respectively. Then Algorithm 1 outputs $\bar{d}_\pi^K$ and $\bar{\lambda}^K$ satisfying $\min_{d_\pi \in \mathcal{K}} \mathcal{L}(d_\pi, \bar{\lambda}^K) \ge f^{OPT} - \epsilon_K - \delta_K$ and $\max_{\lambda \in \Lambda} \mathcal{L}(\bar{d}_\pi^K, \lambda) \le f^{OPT} + \epsilon_K + \delta_K$.*

This theorem tells us that so long as the RL algorithm we employ has guaranteed low-regret, and assuming we choose a reasonable low-regret algorithm for deciding the costs, then the meta-algorithm will produce a solution to the convex MDP problem (Eq. (2)) to any desired tolerance, this is because $f^{OPT} \le f(\bar{d}_\pi^K) = \max_\lambda \mathcal{L}(\bar{d}_\pi^K, \lambda) \le f^{OPT} + \epsilon_K + \delta_K$. For example, we shall later present algorithms that have regret bounded as $\epsilon_K = \delta_K \le O(1/\sqrt{K})$, in which case we have

$$f(\bar{d}_\pi^K) - f^{OPT} \le O(1/\sqrt{K}). \tag{5}$$

**Non-Convex $f$.** Remark 1 implies that the game $\max_{\lambda \in \Lambda} \min_{d_\pi \in \mathcal{K}} (\lambda \cdot d_\pi - f^*(\lambda))$ is concave-convex for any function $f$, so we can solve it with Algorithm 1, even for a non-convex $f$. From weak duality the value of the Lagrangian on the output of Algorithm 1, $\mathcal{L}(\bar{d}_\pi, \bar{\lambda})$, is a lower bound on the optimal solution $f^{OPT}$. In addition, since $f(d_\pi)$ is always an upper bound on $f^{OPT}$ we have both an upper bound and a lower bound on the optimal value: $\mathcal{L}(\bar{d}_\pi, \bar{\lambda}) \le f^{OPT} \le f(\bar{d}_\pi)$.

# 4 Policy and Cost Players for Convex MDPs

In this section we present several algorithms for the policy and cost players that can be used in Algorithm 1. Any combination of these algorithms is valid and will come with different practical and theoretical performance. In Section 6 we show that several well known methods in the literature correspond to particular choices of cost and policy players and so fall under our framework.

In addition, in this section we assume that

$$\lambda_{\max} = \max_{\lambda \in \Lambda} \max_{s,a} |\lambda(s,a)| < \infty,$$

which holds when the set $\Lambda$ is compact. One way to guarantee that $\Lambda$ is compact is to consider functions $f$ with Lipschitz continuous gradients (which implies bounded gradients since the set $\mathcal{K}$ is compact). For simplicity, we further assume that $\lambda_{\max} \leq 1$. By making this assumption we assure that the non stationary rewards produced by the cost player are bounded by 1 as is usually done in RL.

## 4.1 Cost Player

**Follow the Leader (FTL)** is a classic OCO algorithm that selects $\lambda_k$ to be the best point in hindsight. In the special case of convex MDPs, as defined in Eq. (4), FTL has a simpler form:

$$\lambda^k = \arg\max_{\lambda \in \Lambda} \sum_{j=1}^{k-1} \mathcal{L}(d_\pi^j, \lambda) = \arg\max_{\lambda \in \Lambda} \left( \lambda \cdot \sum_{j=1}^{k-1} d_\pi^j - K f^*(\lambda) \right) = \nabla f(\bar{d}_\pi^{k-1}), \quad (6)$$

where $\bar{d}_\pi^{k-1} = \sum_{j=1}^{k-1} d_\pi^j$ and the last equality follows from the fact that $(\nabla f^*)^{-1} = \nabla f$ [56]. The average regret of FTL is guaranteed to be $\bar{R}_K \leq c/\sqrt{K}$ under some assumptions [29]. In some cases, and specifically when the set $\mathcal{K}$ is a polytope and the function $f$ is strongly convex, FTL can enjoy logarithmic or even constant regret; see [32, 29] for more details.

**Online Mirror Descent (OMD)** uses the following update [47, 9]:

$$\lambda^k = \arg\max_{\lambda \in \Lambda} \left( (\lambda - \lambda^{k-1}) \cdot \nabla_\lambda \mathcal{L}(d_\pi^{k-1}, \lambda^{k-1}) + \alpha_k B_r(\lambda, \lambda^{k-1}) \right),$$

where $\alpha_k$ is a learning rate and $B_r$ is a Bregman divergence [14]. For $B_r(x) = 0.5\|x\|_2^2$, we get online gradient descent [79] and for $B_r(x) = x \cdot \log(x)$ we get multiplicative weights [23] as special cases. We also note that OMD is equivalent to a linearized version of Follow the Regularized Leader (FTRL) [43, 28]. The average regret of OMD is $\bar{R}_K \leq c/\sqrt{K}$ under some assumptions, see, for example [28].

## 4.2 Policy Players

### 4.2.1 Best Response

In OCO, the best response is to simply ignore the history and play the best option on the current round, which has guaranteed average regret bound of $\bar{R}_K \leq 0$ (this requires knowledge of the *current* loss function, which is usually not applicable but is in this case). When applied to Eq. (4), it is possible to find the best response $d_\pi^k$ using standard RL techniques since

$$d_\pi^k = \arg\min_{d_\pi \in \mathcal{K}} \mathcal{L}_k(d_\pi, \lambda^k) = \arg\min_{d_\pi \in \mathcal{K}} d_\pi \cdot \lambda^k - f^*(\lambda^k) = \arg\max_{d_\pi \in \mathcal{K}} d_\pi \cdot (-\lambda^k),$$

which is an RL problem for maximizing the reward $(-\lambda^k)$. In principle, any RL algorithm that eventually solves the RL problem can be used to find the best response, which substantiates our claim in the introduction. For example, tabular Q-learning executed for sufficiently long and with a suitable exploration strategy will converge to the optimal policy [72]. In the non-tabular case we could parameterize a deep neural network to represent the Q-values [45] and if the network has sufficient capacity then similar guarantees might hold. We make no claims on efficiency or tractability of this approach, just that in principle such an approach would provide the best-response at each iteration and therefore satisfy the required conditions to solve the convex MDP problem.

### 4.2.2 Approximate Best Response

The caveat in using the best response as a policy player is that in practice, it can only be found approximately by executing an RL algorithm in the environment. This leads to defining an approximate best response via the Probably Approximately Correct (PAC) framework. We say that a policy player is PAC($\epsilon, \delta$), if it finds an $\epsilon$-optimal policy to an RL problem with probability of at least $1 - \delta$. In addition, we say that a policy $\pi'$ is $\epsilon$-optimal if its state occupancy $d'_\pi$ is such that

$$\max_{d_\pi \in \mathcal{K}} d_\pi \cdot (-\lambda^k) - d'_\pi \cdot (-\lambda^k) \leq \epsilon.$$

For example, the algorithm in [40] can find an $\epsilon$-optimal policy to the discounted RL problem after seeing $O\left(\frac{SA}{(1-\gamma)^3 \epsilon^2} \log(\frac{1}{\delta})\right)$ samples; and the algorithm in [36] can find an $\epsilon$-optimal policy for the average reward RL problem after seeing $O\left(\frac{t_{\mathrm{mix}}^2 SA}{\epsilon^2} \log(\frac{1}{\delta})\right)$ samples, where $t_{\mathrm{mix}}$ is the mixing time (see, *eg*, [42, 76] for a formal definition). The following Lemma analyzes the sample complexity of Algorithm 1 with an approximate best response policy player for the average reward RL problem [36]. The result can be easily extended to the discounted case using the algorithm in [40]. Other relaxations to the best response for specific algorithms can be found in [65, 44, 33, 30].

**Lemma 2** (The sample complexity of approximate best response in convex MDPs with average occupancy measure). *For a convex function $f$, running Algorithm 1 with an oracle cost player with regret $\bar{R}_K^\lambda = O(1/K)$ and an approximate best response policy player that solves the average reward RL problem in iteration $k$ to accuracy $\epsilon_k = 1/k$ returns an occupancy measure $\bar{d}_\pi^K$ that satisfies $f(\bar{d}_\pi^K) - f^{OPT} \leq \epsilon$ with probability $1 - \delta$ after seeing $O(t_{mix}^2 SA \log(2K/\epsilon\delta)/\epsilon^3\delta^3)$ samples. Similarly, for $\bar{R}_K^\lambda = O(1/\sqrt{K})$, setting $\epsilon_k = 1/\sqrt{k}$ requires $O(t_{mix}^2 SA \log(2K/\epsilon\delta)/\epsilon^4\delta^4)$ samples.*

### 4.2.3 Non-Stationary RL Algorithms

We now discuss a different type of policy players; instead of solving an MDP to accuracy $\epsilon$, these algorithms perform a *single* RL update to the policy, with cost $-\lambda_k$. In our setup the reward is known and deterministic but non-stationary, while in the standard RL setup it is unknown, stochastic, and stationary. We conjecture that any RL algorithm can be adapted to the *known* non-stationary reward setup we consider here. In most cases both Bayesian [51, 48] and frequentist [8, 35] approaches to the stochastic RL problem solve a modified (*eg*, by adding optimism) Bellman equation at each time period and swapping in a known but non-stationary reward is unlikely to present a problem.

To support this conjecture we shall prove that this is exactly the case for UCRL2 [35]. UCRL2 is an RL algorithm that was designed and analyzed in the standard RL setup, and we shall show that it is easily adapted to the non-stationary but known reward setup that we require. To make this claim more general, we will also discuss a similar result for the MDPO algorithm [61] that was given in a slightly different setup.

UCRL2 is a model based algorithm that maintains an estimate of the reward and the transition function as well as confidence sets about those estimates. In our case the reward at time $k$ is known, so we only need to consider uncertainty in the dynamics. UCRL2 guarantees that in any iteration $k$, the true transition function is in a confidence set with high probability, *i.e.*, $P \in \mathcal{P}_k$ for confidence set $\mathcal{P}_k$. If we denote by $J_\pi^{P,R}$ the value of policy $\pi$ in an MDP with dynamics $P$ and reward $R$ then the optimistic policy is $\tilde{\pi}_k = \arg\max_\pi \max_{P' \in \mathcal{P}_k} J_\pi^{P', -\lambda_k}$. Acting according to this policy is guaranteed to attain low regret. In the following results for UCRL2 we will use the constant $D$, which denotes the diameter of the MDP, see [35, Definition 1] for more details. In the supplementary material (Appendix E), we provide a proof sketch that closely follows [35].

**Lemma 3** (Non stationary regret of UCRL2). *For an MDP with dynamics $P$, diameter $D$, an arbitrary sequence of known and bounded rewards $\left\{r^i : \max_{s,a} |r^i(s,a)| \leq 1\right\}_{i=1}^K$, such that the optimal average reward at episode $k$, with respect to $P$ and $r_k$ is $J_k^\star$, then with probability at least $1 - \delta$, the average regret of UCRL2 is at most $\bar{R}_K = \frac{1}{K} \sum_{k=1}^K J_k^\star - J_k^{\tilde{\pi}_k} \leq O(DS\sqrt{A \log(K/\delta)/K})$.*

Next, we give a PAC($\epsilon, \delta$) sample complexity result for the mixed policy $\bar{\pi}^K$, that is produced by running Algorithm 1 with UCRL2 as a policy player.

**Lemma 4** (The sample complexity of non-stationary RL algorithms in convex MDPs). *For a convex function $f$, running Algorithm 1 with an oracle cost player with regret $\bar{R}_K^\lambda \leq c_0/\sqrt{K}$ and UCRL2 as a policy player returns an occupancy measure $\bar{d}_\pi^K$ that satisfies $f(\bar{d}_\pi^K) - f^{OPT} \leq \epsilon$ with probability $1 - \delta$ after $K = O\left(\frac{D^2 S^2 A}{\delta^2 \epsilon^2} \log(\frac{2DSA}{\delta\epsilon})\right)$ steps.*

**MDPO.** Another optimistic algorithm is Mirror Descent Policy Optimization [60, MDPO]. MDPO is a model free RL algorithm that is very similar to popular DRL algorithms like TRPO [58] and MPO [2]. In [24, 59, 5], the authors established the global convergence of MDPO and in [15, 60], the authors showed that MDPO with optimistic exploration enjoys low regret.

The analysis for MDPO is given in a finite horizon MDP with horizon $H$, which is not the focus of our paper. Nevertheless, to support our conjecture that any stochastic RL algorithm can be adapted to the *known* non-stationary reward setup, we quickly discuss the regret of MDPO in this setup. We also note that MDPO is closer to practical DRL algorithms [70]. In a finite horizon MDP with horizon $H$ and known, non-stationary and bounded rewards, the regret of MDPO is bounded by $\bar{R}_K \leq O(H^2 S \sqrt{A/K})$ [61, Lemma 4] with high probability.

To compare this result with UCRL2, we refer to a result from [57], which analyzed UCRL2 in the adversarial setup, that includes our setup as a special case. In a finite horizon MDP with horizon $H$ it was shown that setting $\delta = SA/K$ with probability $1 - \delta$ its regret is bounded by $\bar{R}_K \leq O(HS\sqrt{A\log(K)/K})$ [57, Corollary 5], which is better by a factor of $H$ than MDPO.

**Discussion.** Comparing the results in Lemma 4 with Lemma 2 suggests that using an RL algorithm with non stationary reward as a policy player requires $O(1/\epsilon^2)$ samples to find an $\epsilon-$optimal policy, while using an approximate best response requires $O(1/\epsilon^3)$. In first glance, this results also improves the previously best known result of Hazan et al. [30] for approximate Frank-Wolfe (FW) that requires $O(1/\epsilon^3)$ samples. However, there are more details that have to be considered as we now discuss.

Firstly, Lemma 4 and Lemma 2 assume access to an oracle cost player with some regret and do not consider how to implement such a cost player. The main challenge is that the cost player does not have access to the true state occupancy and must estimate it from samples. If we do not reuse samples from previous policies to estimate the state occupancy of the current policy we will require $O(1/\epsilon^3)$ trajectories overall [30]. A better approach would use the samples from previous episodes to learn the transition function. Then, given the estimated transition function and the policy, we can compute an approximation of the state occupancy. We conjecture that such an approach would lead to a $O(1/\epsilon^2)$ sample complexity, closing the gap with standard RL.

Secondly, while our focus is on the dependence in $\epsilon$, our bound Lemma 4 is not tight in $\delta$, *i.e.*, it scales with $1/\delta^2$ where it should be possible to achieve a $\log(1/\delta)$ scaling. Again we conjecture an improvement in the bound is possible; see, *eg*, [38, Appendix F.].

## 5   Convex Constraints

We have restricted the presentation so far to unconstrained convex problems, in this section we extend the above results to the constrained case. The problem we consider is

$$\min_{d_\pi \in \mathcal{K}} f(d_\pi) \quad \text{subject to} \quad g_i(d_\pi) \leq 0, \quad i = 1, \ldots m,$$

where $f$ and the constraint functions $g_i$ are convex. Previous work focused on the case where both $f$ and $g_i$ are linear [7, 67, 12, 68, 18, 16, 11]. We can use the same Fenchel dual machinery we developed before, but now taking into account the constraints. Consider the Lagrangian

$$L(d_\pi, \mu) = f(d_\pi) + \sum_{i=1}^m \mu_i g_i(d_\pi) = \max_\nu \left(\nu \cdot d_\pi - f^*(\nu)\right) + \sum_{i=1}^m \mu_i \max_{v_i} \left(d_\pi v_i - g_i^*(v_i)\right).$$

over dual variables $\mu \geq 0$, with new variables $v_i$ and $\nu$. At first glance this does not look convex-concave, however we can introduce new variables $\zeta_i = \mu_i v_i$ to obtain

$$L(d_\pi, \mu, \nu, \zeta_1, \ldots, \zeta_m) = \nu \cdot d_\pi - f^*(\nu) + \sum_{i=1}^m \left(d_\pi \zeta_i - \mu_i g_i^*(\zeta_i/\mu_i)\right). \tag{7}$$

This is convex (indeed affine) in $d_\pi$ and concave in $(\nu, \mu, \zeta_1, \ldots, \zeta_m)$, since it includes the perspective transform of the functions $g_i$ [13]. The Lagrangian involves a cost vector, $\nu + \sum_{i=1}^m \zeta_i$, linearly

interacting with $d_\pi$, and therefore we can use the same policy players as before to minimize this cost. For the cost player, it is possible to use OMD on Eq. (7) jointly for the variables $\nu, \mu$ and $\zeta$. It is more challenging to use best-response and FTL for the cost-player variables as the maximum value of the Lagrangian is unbounded for some values of $d_\pi$. Another option is to treat the problem as a *three*-player game. In this case the policy player controls $d_\pi$ as before, one cost player chooses $(\nu, \zeta_1, \ldots, \zeta_m)$ and can use the algorithms we have previously discussed, and the other cost player chooses $\mu$ with some restrictions on their choice of algorithm. Analyzing the regret in that case is outside the scope of this paper.

## 6  Examples

In this section we explain how existing algorithms can be seen as instances of the meta-algorithm for various choices of the objective function $f$ and the cost and policy player algorithms $\text{Alg}_\lambda$ and $\text{Alg}_\pi$. We summarized the relationships in Table 1.

### 6.1  Apprenticeship Learning

In apprenticeship learning (AL), we have an MDP without an explicit reward function. Instead, an expert provides demonstrations which are used to estimate the expert state occupancy measure $d_E$. Abbeel and Ng [1] formalized the AL problem as finding a policy $\pi$ whose state occupancy is close to that of the expert by minimizing the convex function $f(d_\pi) = ||d_\pi - d_E||$. The convex conjugate of $f$ is given by $f^*(y) = y \cdot d_E$ if $||y||_* \leq 1$ and $\infty$ otherwise, where $||\cdot||_*$ denotes the dual norm. Plugging $f^*$ into Eq. (4) results in the following game:

$$\min_{d_\pi \in \mathcal{K}} ||d_\pi - d_E|| = \min_{d_\pi \in \mathcal{K}} \max_{||\lambda||_* \leq 1} \lambda \cdot d_\pi - \lambda \cdot d_E. \tag{8}$$

Inspecting Eq. (8), we can see that the norm in the function $f$ that is used to measure the distance from the expert induces a constraint set for the cost variable, which is a unit ball in the dual norm.

**$\text{Alg}_\lambda$=OMD, $\text{Alg}_\pi$=Best Response/RL.**  The Multiplicative Weights AL algorithms [65, MWAL] was proposed to solve the AL problem with $f(d_\pi) = ||d_\pi - d_E||_\infty$. It uses the best response as the policy player and multiplicative weights as the cost player (a special case of OMD). MWAL has also been used to solve AL in contextual MDPs [10] and to find feasible solutions to convex-constrained MDPs [44]. We note that in practice the best response can only be solved approximately, as we discussed in Section 4. Instead, in online AL [61] the authors proposed to use MDPO as the policy player, which guarantees a regret bound of $\bar{R}_K \leq c/\sqrt{K}$. They showed that their algorithm is equivalent to Wasserstein GAIL [73, 78] and in practice tends to perform similarly to GAIL.

**$\text{Alg}_\lambda$=FTL, $\text{Alg}_\pi$=Best Response.**  When the policy player plays the best response and the cost player plays FTL, Algorithm 1 is equivalent to the Frank-Wolfe algorithm [22, 3] for minimizing $f$ (Eq. (2)). Pseudo-code for this is included in the appendix (Algorithm 3). The algorithm finds a point $d_\pi^k \in \mathcal{K}$ that has the largest inner-product (best response) with the negative gradient (*i.e.*, FTL).

Abbeel and Ng [1] proposed two algorithms for AL, the projection algorithm and the max margin algorithm. The projection algorithm is essentially a FW algorithm, as was suggested in the supplementary [1] and was later shown formally in [75]. Thus, it is a projection free algorithm in the sense that it avoids projecting $d_\pi$ into $\mathcal{K}$, despite the name. In their case the gradient is given by $\nabla_f(d_\pi) = d_\pi - d_E$. Thus, finding the best response is equivalent to solving an MDP whose reward is $d_E - d_\pi$. In a similar fashion, FW can be used to solve convex MDPs more generally [30]. Specifically, in [30], the authors considered the problem of pure exploration, which they defined as finding a policy that maximizes entropy.

**Fully Corrective FW.** The FW algorithm has many variants (see [33] for a survey) some of which enjoy faster rates of convergence in special cases. Concretely, when the constraint set is a polytope, which is the case for convex MDPs (Definition 1), some variants achieve a linear rate of convergence [34, 75]. One such variant is the Fully corrective FW, which replaces the learning rate update (see line 4 of Algorithm 3 in the supplementary), with a minimization problem over the convex hull of occupancy measures at the previous time-step. This is guaranteed to be at least as good as the learning rate update. Interestingly, the second algorithm of Abbeel and Ng [1], the max margin algorithm, is

exactly equivalent to this fully corrective FW variant. This implies that the max-margin algorithm enjoys a better theoretical convergence rate than the 'projection' variant, as was observed empirically in [1].

## 6.2 GAIL and DIAYN: $\text{Alg}_\lambda$=FTL, $\text{Alg}_\pi$=RL

We now discuss the objectives of two popular algorithms, GAIL [31] and DIAYN [20], which perform AL and diverse skill discovery respectively. Our analysis suggests that GAIL and DIAYN share the same objective function. In GAIL, this objective function is minimized, which is a convex MDP, however, in DIAYN it is maximized, which is therefore not a convex MDP. We start the discussion with DIAYN and follow with a simple construction showing the equivalence to GAIL.

**DIAYN.** Discriminative approaches [26, 20] rely on the intuition that skills are diverse when they are entropic and easily discriminated by observing the states that they visit. Given a probability space $(\Omega, \mathcal{F}, \mathcal{P})$, state random variables $S : \Omega \to \mathcal{S}$ and latent skills $Z : \Omega \to \mathcal{Z}$ with prior $p$, the key term of interest being maximized in DIAYN [20] is the mutual information:

$$I(S; Z) = \mathbb{E}_{z \sim p; s \sim d_\pi^z}[\log p(z|s) - \log p(z)], \tag{9}$$

where $d_\pi^z$ is the stationary distribution induced by the policy $\pi(a \mid s, z)$. For each skill $z$, this corresponds to a standard RL problem with (conditional) policy $\pi(a \mid s, z)$ and reward function $r(s|z) = \log p(z|s) - \log p(z)$. The first term encourages the policy to visit states for which the underlying skill has high-probability under the posterior $p(z \mid s)$, while the second term ensures a high entropy distribution over skills. In practice, the full DIAYN objective further regularizes the learnt policy by including entropy terms $-\log \pi(a \mid s, z)$. For large state spaces, $p(z|s)$ is typically intractable and Eq. 9 is replaced with a variational lower-bound, where the true posterior is replaced with a learned discriminator $q_\phi(z|s)$. Here, we focus on the simple setting where $z$ is a categorical distribution over $|Z|$ outcomes, yielding $|Z|$ policies $\pi^z$, and $q_\phi$ is a classifier over these $|Z|$ skills with parameters $\phi$.

We now show that a similar intrinsic reward can be derived using the framework of convex MDPs. We start by writing the true posterior as a function of the per-skill state occupancy $d_\pi^z = p(s \mid z)$, and using Bayes rules, $p(z|s) = \frac{d_\pi^z(s)p(z)}{\sum_k d_\pi^k(s)p(k)}$. Combing this with Eq. (9) yields:

$$\mathbb{E}_{z \sim p(z), s \sim d_\pi^z}[\log p(z|s) - p(z)] = \sum_z p(z) \sum_s d_\pi^z(s) \left[\log\left(\frac{d_\pi^z(s)p(z)}{\sum_k d_\pi^k(s)p(k)}\right) - \log p(z)\right]$$

$$= \sum_z p(z)\text{KL}(d_\pi^z || \sum_k p(k)d_\pi^k) = \mathbb{E}_z \text{KL}(d_\pi^z || \mathbb{E}_k d_\pi^k), \tag{10}$$

where KL denotes the Kullback–Leibler divergence [39].

Intuitively, finding a set of policies $\pi^1, \ldots, \pi^z$ that minimize Eq. (10) will result in finding policies that visit similar states, measured using the KL distance between their respective state occupancies $d_\pi^1, \ldots, d_\pi^z$. This is a convex MDP because the KL-divergence is jointly convex in both arguments [13, Example 3.19]. We will soon show that this is the objective of GAIL. On the other hand, a set of policies that maximize Eq. (10) is diverse, as the policies visit different states, measured using the KL distance between their respective state occupancies $d_\pi^1, \ldots, d_\pi^z$.

We follow on with deriving the FTL player for the convex MDP in Eq. (10). We will then show that this FTL player is producing an intrinsic reward that is equivalent to the intrinsic reward used in GAIL and DIAYN (despite the fact that DIAYN is not a convex MDP). According to Eq. (6), the FTL cost player will produce a cost $\lambda^k$ at iteration $k$ given by

$$\nabla_{d_\pi^z}\text{KL}(d_\pi^z || \sum_k p(k)d_\pi^k) = \mathbb{E}_{z \sim p(z)}\left[\log \frac{d_\pi^z}{\sum_k d_\pi^k p(k)} + 1 - \frac{d_\pi^z p(z)}{\sum_k d_\pi^k p(k)}\right]$$

$$= \mathbb{E}_{z \sim p(z)}\Big[\underbrace{\log(p(z|s)) - \log(p(z))}_{\text{Mutual Information}} \underbrace{+1 - p(z|s)}_{\text{Gradient correction}}\Big], \tag{11}$$

where the equality follows from writing the posterior as a function of the per-skill state occupancy $d_\pi^z = p(s \mid z)$, and using Bayes rules, $p(z|s) = \frac{d_\pi^z(s)p(z)}{\sum_k d_\pi^k(s)p(k)}$. Replacing the posterior $p(z|s)$ with

a learnt discriminator $q_\phi(z|s)$ recovers the mutual-information rewards of DIAYN, with additional terms $1 - p(z \mid s)$ which we refer to as "gradient correction" terms. Inspecting the common scenario of a uniform prior over the latent variables, $p(z) = 1/|Z|$, we get that the expectation of the gradient correction term $\sum_z p(z)(1 - p(z|s)) = 1 - 1/|Z|$ in each state. From the perspective of the policy player, adding a constant to the reward does not change the best response policy, nor the optimistic policy. Therefore, the gradient correction term does not have an effect on the optimization under a uniform prior, and we retrieved the reward of DIAYN. These algorithms differ however for more general priors $p(z)$, which we explore empirically in Appendix F.

**GAIL.** We further show how Eq. (10) extends to GAIL [31] via a simple construction. Consider a binary latent space of size $|Z| = 2$, where $z = 1$ corresponds to the policy of the agent and $z = 2$ corresponds to the policy of the expert which is fixed. In addition, consider a uniform prior over the latent variables, *i.e.*, $p(z = 1) = \frac{1}{2}$. By removing the constant terms in Eq. (11), one retrieves the GAIL [31] algorithm. The cost $\log(p(z|s))$ is the probability of the discriminator to identify the agent, and the policy player is MDPO (which is similar to TRPO in GAIL).

# 7 Discussion

In this work we reformulated the convex MDP problem as a convex-concave game between the agent and another player that is producing costs (negative rewards) and proposed a meta-algorithm for solving it.

We observed that many algorithms in the literature can be interpreted as instances of the meta-algorithm by selecting different pairs of subroutines employed by the policy and cost players. The Frank-Wolfe algorithm, which combines best response with FTL, was originally proposed for AL [1, 75] but can be used for any convex MDP problem as was suggested in [30]. Zhang et al. [77], unified the problems of RL, AL, constrained MDPs with linear constraints and maximum entropy exploration under the framework of convex MDPs. We extended the framework to allow convex constraints (Section 5) and explained the objective of GAIL as a convex MDP (Section 6.2). We also discussed non convex objectives (Section 3) and analyzed unsupervised skill discovery via the maximization of mutual information (Section 6.2) as a special case. Finally, we would like to point out a recent work by Geist et al. [25], which was published concurrently to ours, and studies the convex MDP problem from the viewpoint of mean field games.

There are also algorithms for convex MDPs that cannot be explained as instances of Algorithm 1. In particular, Zhang et al. [77] proposed a policy gradient algorithm for convex MDPs in which each step of policy gradient involves solving a new saddle point problem (formulated using the Fenchel dual). This is different from our approach since we solve a single saddle point problem iteratively, and furthermore we have much more flexibility about which algorithms the policy player can use. Moreover, for the convergence guarantee [77, Theorem 4.5] to hold, the saddle point problem has to be solved exactly, while in practice it is only solved approximately [77, Algorithm 1], which hinders its sample efficiency. Fenchel duality has also been used in off policy evaluation (OPE) in [46, 74]. The difference between these works and ours is that we train a policy to minimize an objective, while in OPE a target policy is fixed and its value is estimated from data produced by a behaviour policy.

In order to solve a practical convex MDP problem in a given domain it would be prudent to use an RL algorithm that is known to be high performing for the vanilla RL problem as the policy player. From the theoretical point of view this could be MDPO or UCRL2, which we have shown come with strong guarantees. From the practical point of view using a high performing DRL algorithm, which may be specific to the domain, will usually yield the best results. For the cost player using FTL, *i.e.*, using the gradient of the objective function, is typically the best choice.

## Acknowledgments and Disclosure of Funding

We would like to thank Yasin Abbasi-Yadkorie, Vlad Mnih, Jacob Abernethy, Lior Shani and Doina Precup for their comments and discussion on this work. Work done at DeepMind, the authors received no specific funding for this work.

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
