&= (1-\gamma)\sum_{t=1}^{\infty}\sum_s \mathbb{P}_\pi(s_t = s)\sum_a \pi(s,a)\gamma^t r(s,a) \\
&= (1-\gamma)\sum_{s,a}\left(\sum_{t=1}^{\infty}\gamma^t \mathbb{P}_\pi(s_t = s)\pi(s,a)\right) r(s,a) \\
&= \sum_{s,a} d_\pi^\gamma(s,a) r(s,a).
\end{aligned}
$$

Similarly, for the average reward case

$$
\begin{aligned}
J_\pi^{\mathrm{avg}} &= \lim_{T\to\infty}\frac{1}{T}\mathbb{E}\sum_{t=1}^{T} r_t \\
&= \lim_{T\to\infty}\frac{1}{T}\sum_{t=1}^{T}\sum_s \mathbb{P}_\pi(s_t = s)\sum_a \pi(s,a) r(s,a) \\
&= \sum_{s,a}\left(\lim_{T\to\infty}\frac{1}{T}\sum_{t=1}^{T}\mathbb{P}_\pi(s_t = s)\pi(s,a)\right) r(s,a) \\
&= \sum_{s,a} d_\pi^{\mathrm{avg}}(s,a) r(s,a).
\end{aligned}
$$

∎

## C FW algorithms

### C.1 Pseudo code

---
**Algorithm 3:** Frank-Wolfe algorithm

---
    **Input:** a convex and smooth function $f$
2:  **Initialize:** Pick a random element $d_\pi^1 \in \mathcal{K}$.
    **for** $i = 1, \ldots, T$ **do**
4:    $d_\pi^{k+1} = \arg\max_{\pi \in \Pi} d_\pi \cdot -\nabla f(\bar{d}_\pi^k)$
     $\bar{d}_\pi^{k+1} = (1 - \alpha_i)\bar{d}_\pi^k + \alpha_i d_\pi^{k+1}$
6:  **end for**

---

### C.2 Linear convergence

**Theorem 2** (Linear Convergence [34]). *Suppose that f has L-Lipschitz gradient and is $\mu$-strongly convex. Let $D = \{d_\pi, \forall \pi \in \Pi\}$ be the set of all the state occupancy's of deterministic policies in the MDP and let $\mathcal{K} = Co(D)$ be its Convex Hull. Such that $\mathcal{K}$ a polytope with vertices $D$, and let $M = diam(\mathcal{K})$. Also, denote the Pyramidal Width of $D$, $\delta = PWidth(D)$ as in [34, Equation 9 1].*

*Then the suboptimality $h_t$ of the iterates of all the fully corrective FW algorithm decreases geometrically at each step, that is*

$$f(\bar{d}_\pi^{k+1}) \leq (1 - \rho)f(\bar{d}_\pi^k), \text{ where } \rho = \frac{\mu\delta^2}{4LM^2}$$

## D Sample complexity proofs

**Lemma** (The sample complexity of non-stationary RL algorithms in convex MDPs). *For a convex function $f$, running Algorithm 1 with an oracle cost player with regret $\bar{R}_K^\lambda \leq c_0/\sqrt{K}$ and UCRL2 as a policy player returns a mixed policy $\bar{\pi}^K$ that satisfies $f(\bar{d}_\pi^K) - f^{OPT} \leq \epsilon$ with probability $1 - \delta$ after $K = O\left(\frac{D^2 S^2 A}{\delta^2 \epsilon^2} \log(\frac{2DSA}{\delta\epsilon})\right)$ steps.*

**Proof**. In Theorem 1 and the discussion below it, we showed that $f(\bar{d}_\pi^K) - f^{OPT} \leq \bar{R}_K^\lambda + \bar{R}_K^\pi$. From Lemma 3, we have that with probability $1 - \delta'$, a "positive event" happens, and the regret of the UCRL2 player, $\epsilon_K$, is upper bounded by $\bar{R}_K^\pi = \frac{1}{K}\sum_{k=1}^K J_k^\star - J_k^{\tilde{\pi}_k} \leq c_1 DS\sqrt{A\log(K/\delta')/K}$ for some constant $c_1$. Recall that the function $f$ has bounded gradients and therefore, the non stationary reward is upper bounded by 1. Thus, when the "positive event" does not happen (with probability $\delta'$), we can always upper bound the regret by $\bar{R}_K^\pi = \frac{1}{K}\sum_{k=1}^K J_k^\star - J_k^{\tilde{\pi}_k} \leq 1$. Using Markov's inequality, we have that

$$\Pr(f(\bar{d}_\pi^K) - f^{OPT} \geq \epsilon) \leq \frac{\mathbb{E}f(\bar{d}_\pi^K) - f^{OPT}}{\epsilon} \leq \frac{\mathbb{E}(\bar{R}_K^\pi + \bar{R}_K^\lambda)}{\epsilon}$$
$$\leq \frac{1}{\epsilon}\left((1 - \delta')c_1 DS\sqrt{A\log(K/\delta')/K} + \delta' + c_0\sqrt{1/K}\right)$$
$$\leq \frac{1}{\epsilon}\left((c_2 DS\sqrt{A\log(K/\delta')/K}) + \delta'\right),$$

thus, if we choose $\delta' = \frac{\epsilon\delta}{2}$, we have that in order for $\Pr(f(\bar{d}_\pi^K) - f^{OPT} \geq \epsilon)$ to be smaller than $\delta$ after $K$ steps, it is enough to find a value for $K$ such that $\frac{1}{\epsilon}c_2 DS\sqrt{A\log(2K/\epsilon\delta)/K} \leq \delta/2$, which is achieved for $K = \frac{cD^2 S^2 A}{\delta^2 \epsilon^2}\log(\frac{2DSA}{\delta\epsilon})$   ■.

**Lemma** (The sample complexity of approximate best response in convex MDPs with average occupancy measure). *For a convex function $f$, running Algorithm 1 with an oracle cost player with regret $\bar{R}_K^\lambda = O(1/K)$ and an approximate best response policy player that solves the average reward RL problem in iteration $k$ to accuracy $\epsilon_k = 1/k$ returns an occupancy measure $\bar{d}_\pi^K$ that satisfies $f(\bar{d}_\pi^K) - f^{OPT} \leq \epsilon$ with probability $1 - \delta$ after seeing $O(t_{mix}^2 SA\log(2K/\epsilon\delta)/\epsilon^3\delta^3)$ samples. Similarly, for $\bar{R}_K^\lambda = O(1/\sqrt{K})$, setting $\epsilon_k = 1/\sqrt{k}$ requires $O(t_{mix}^2 SA\log(2K/\epsilon\delta)/\epsilon^4\delta^4)$ samples.*

To solve an MDP to accuracy $\epsilon_k$, it is sufficient to run an RL algorithm for $O(1/\epsilon_k^2)$ iterations. This is a lower bound and an upper bound in $\epsilon$, see, for example [36] for an upper bound of $O\left(\frac{t_{\text{mix}}^2 SA}{\epsilon^2} \log(1/\delta)\right)$ and a lower bound [37] of $O\left(\frac{t_{\text{mix}} SA}{\epsilon^2} \log(1/\delta)\right)$ for the average reward case.

We continue the proof using the algorithm of [17] as the approximate best response player, *i.e.*, we invoke their algorithm at iteration $k$ to find an $\epsilon_k-$optimal solution with probability $1-\delta_k = 1-\delta'/K$. Applying the union bound over the iterations gives us that with probability of $1 - \delta'$, the regret of the policy player is $\bar{R}_K^\pi = \frac{1}{K} \sum \epsilon_k$. [1]

We consider two cases for the cost player. In the first, we will consider average regret of $\bar{R}_K^\lambda = c/\sqrt{K}$, which is feasible for any of the cost players we considered in this paper. In this case, we will set the per-iteration $\epsilon$ to be $\epsilon_k = c/\sqrt{k}$. We have that

$$\bar{R}_K = \bar{R}_K^\pi + \bar{R}_K^\lambda \leq \frac{1}{K} \sum_{k=1}^K c_2/\sqrt{k} + c_1/\sqrt{K} \leq \frac{1}{K} c_3\sqrt{k} + c_1/\sqrt{K} \leq c_4/\sqrt{K}.$$

Then, via Markov inequality we get that

$$\Pr(f(\bar{d}_\pi^K) - f^{\text{OPT}} \leq \epsilon) \leq \frac{\mathbb{E}f(\bar{d}_\pi^K) - f^{\text{OPT}}}{\epsilon} \leq \frac{\mathbb{E}\bar{R}_K^\pi + \bar{R}_K^\lambda}{\epsilon} \leq \frac{1}{\epsilon}\left(\frac{c_4}{\sqrt{K}} + \delta'\right).$$

Setting $\delta' = \epsilon\delta/2$ implies that it is enough to run the algorithm for $K = c_5/\epsilon^2\delta^2$ iterations to find an $\epsilon-$optimal solution with probability of $1 - \delta$.

In each iteration $k$, in order to find an $\epsilon_k = c_1/\sqrt{k}$ optimal solution w.p $1 - \delta'/K$, we need to collect $c_2 k t_{\text{mix}}^2 SA \log(K/\delta')$ samples. Thus, the total number of samples is

$$\sum_{k=1}^{c_5/\epsilon^2\delta'^2} c_2 k t_{\text{mix}}^2 SA \log(K/\delta') = c_2 t_{\text{mix}}^2 SA \log(K/\delta') \sum_{k=1}^{c_5/\epsilon^2\delta^2} k \leq c_3 t_{\text{mix}}^2 SA \log(2K/\epsilon\delta)/\epsilon^4\delta^4.$$

In the second scenario we have a cost player with constant regret, and therefore average regret of $\bar{R}_K^\lambda \leq c_1/K$, which is possible to achieve under some assumptions [32]. We set $\epsilon_k = c/k$ and get that

$$\bar{R}_K = \bar{R}_K^\pi + \bar{R}_K^\lambda \leq \frac{1}{K} \sum_{k=1}^K c_2/k + c_1/K \leq \frac{\log(K)}{K} c_3\sqrt{k} + c_1/K \leq c_4\frac{\log(K)}{K}.$$

Then, via Markov inequality we get that

$$\Pr(f(\bar{d}_\pi^K) - f^{\text{OPT}} \leq \epsilon) \leq \frac{\mathbb{E}f(\bar{d}_\pi^K) - f^{\text{OPT}}}{\epsilon} \leq \frac{\mathbb{E}\bar{R}_K^\pi + \bar{R}_K^\lambda}{\epsilon} \leq \frac{1}{\epsilon}\left(\frac{c_4 \log(K)}{K} + \delta'\right).$$

Setting $\delta' = \epsilon\delta/2$ implies that it is enough to run the algorithm for $K = c_5/\epsilon\delta$ iterations to find an $\epsilon-$optimal solution with probability of $1 - \delta$.

In each iteration $k$, in order to find an $\epsilon_k = c_1/k$ optimal solution w.p $1 - \delta'/K$, we need to collect $c_2 k^2 t_{\text{mix}}^2 SA \log(K/\delta')$ samples, which leads to a total number of samples of

$$\sum_{k=1}^{c_5/\epsilon\delta} c_2 k^2 t_{\text{mix}}^2 SA \log(K/\delta') = c_2 t_{\text{mix}}^2 SA \log(K/\delta') \sum_{k=1}^{c_5/\epsilon\delta} k^2 \leq c_3 t_{\text{mix}}^2 SA \log(2K/\epsilon\delta)/\epsilon^3\delta^3. \quad \blacksquare$$

**Discussion on how to choose the schedule for $\epsilon_k$**

The overall regret of the game is the sum of the regret of the policy player and the cost player, and the regret of the game is asymptotically

$$\bar{R}_K = \bar{R}_K^\pi + \bar{R}_K^\lambda = O\left(\max(\bar{R}_K^\pi, \bar{R}_K^\lambda)\right) \tag{12}$$

---

[1]Note that the iterations are independent from each other from the perspective of the approximate best response player so it is possible to apply the union bound. This is because each iteration involves solving a new MDP, and the upper bound does not make any assumptions about the structure of the reward in this MDP.

Consider the general case of $\epsilon_k = 1/k^p$. Note that for the average regret $\frac{1}{K}\sum_{k=1}^K 1/k^p$ to go to zero as $K$ grows, the sum $\frac{1}{K}\sum_{k=1}^K 1/k^p$ must be smaller than $K$, so $p$ must be positive. In addition, for larger values of $p$, $\epsilon_k$ is smaller. Thus the regret is smaller, but at the same time, it requires more samples to solve each RL problem. Inspecting the maximum in Eq. (12), we observe that it does not make sense to choose a value for $p$ for which $\frac{1}{K}\sum_{k=1}^K 1/k^p < \bar{R}_K^\lambda$, since it will not improve the overall regret and will require more samples, than, for example, setting $p$ such that $\frac{1}{K}\sum_{k=1}^K 1/k^p = \bar{R}_K^\lambda$.

Thus, in the case that the cost player has constant regret, $\bar{R}_K^\lambda = O(1/K)$, we set $p \in (0,1]$, and in the case that the cost player has regret of $\bar{R}_K^\lambda = O(1/\sqrt{K})$, we set $p \in (0,0.5]$.

We now continue and further inspect the regret. We have that $\frac{1}{K}\sum_{k=1}^K \epsilon_k = \frac{1}{K}\sum_{k=1}^K 1/k^p = O(k^{-p})$ for $p \in (0,1)$, and $\log(K)/K$ for $p = 1$. Neglecting logarithmic terms, we continued with $O(k^{-p})$ for both cases. In other words, it is sufficient to run the meta-algorithm for $K = 1/\epsilon^p$ iterations to guarantee an error of at most $\epsilon$ for the convex MDP problem.

Thus, to solve an MDP to accuracy $\epsilon_k = 1/k^p$ it requires $k^{2p}$ iterations, and the overall sample complexity is therefore $\sum_{k=1}^{1/\epsilon^p} k^{2p} = O(1/\epsilon^{\frac{2p+1}{p}})$.

The function $1/\epsilon^{\frac{2p+1}{p}}$ is monotonically increasing in $p$, so it attains minimum for the highest value of $p$ which is 0.5 or 1, depending on the cost player. We conclude that the optimal sample complexity with approximate best response is $O(1/\epsilon^3)$ for the cost player that has constant regret and $O(1/\epsilon^4)$ for a cost player with average regret of $\bar{R}_K^\lambda = O(1/\sqrt{K})$.

## E   Proof sketch for Lemma 3

We denote by $r_k^*$ the optimal average reward at time $k$ in an MDP with dynamics $P$ and reward $r_k = -\lambda_k$. We want to show that

$$R_k = \sum_k r_k^* - r_k(s_k, a_k) \leq c/\sqrt{K},$$

that is, that the total reward that the agent collects has low regret compared to the sum of optimal average rewards.

To show that, we make two minor adaptations to the UCRL2 algorithm and then verify that its original analysis also applies to this non-stationary setup. The first modification is that the nonstatioanry version of UCRL2 uses the known reward $r_k$ at time $k$ (which in our case is the output of the cost player) instead of estimating the unknown, stochastic, stationary, extrinsic reward. Since the current reward $r_k$ is known and deterministic, there is no uncertainty about it, and we only have to deal with uncertainty with respect to to the dynamics. The second modification is that we compute a new optimistic policy (using extended value iteration) in each iteration. This optimistic policy is computed with the current reward $r_k$, and the current uncertainty set about the dynamics $\mathcal{P}_k$. This also means that all of our episodes are of length 1.

After making these two clarifications, we follow the proof of UCRL2 and make changes when appropriate. We note that the analysis, basically, does not require any modifications, but we repeat the relevant parts for completeness. We begin with the definition of the regret at episode $k$, which is now just the regret at time $k$ :

$$\Delta_k = \sum_{s,a} v_k(s,a)(r_k^* - r_k(s,a)),$$

where $v_k(s,a)$ in our case is an indicator on the state action pair $s_k, a_k$, and $R_k = \sum_k \Delta_k$.

The instantaneous regret $\Delta_k$ measures the difference between the optimal average reward $r_k^*$, with respect to reward $r_k$, and the reward $r_k(s,a)$ that the agent collected at time $k$ by visiting state $s$ and taking action $k$ from the reward that is produced by the cost player.

Section 4.1 in the UCRL2 paper is the first step in the analysis. It bounds possible fluctuations in the random reward. This step is not required in our case since our reward at time $k$ is the output of the cost player, which is known in all the states and deterministic.

Section 4.2 considers the regret that is caused by failing confidence regions, that is, the event that the true dynamics and true reward are not in the confidence region. In our case there is only confidence region for the dynamics (since the reward is known), which we denote by $\mathcal{P}_k$. Summing the expected regret from episodes in which $P \notin \mathcal{P}_k$ results in a $\sqrt{K}$ term in the regret,

$$\Delta_k \leq \sum_{s,a} v_k(s,a)(r_k^* - r_k(s,a)) + \sqrt{K},$$

where from now on, we continue with the event that $P \in \mathcal{P}_k$.

Next, we denote the optimistic policy and optimistic MDP as the solution of the following problem $\tilde{\pi}_k, \tilde{P}_k = \arg\max_{\pi \in \Pi, P' \in \mathcal{P}_k} J_\pi^{P', r_k}$. In addition, we denote by $\tilde{r}_k$ the optimstic average reward, that is, the average reward of the policy $\tilde{\pi}_k$ in the MDP with the optimstic dynamics $\tilde{P}_k$ and reward $r_k$. We also note that $\tilde{\pi}_k$ is the optimal average reward policy in this MDP by its definition.

We now continue with the case that $P \in \mathcal{P}_k$. The next step is to bound the difference between the optimal average reward $r_k^*$ and the optimistic average reward $\tilde{r}_k$. We note that both $\tilde{r}_k$ and $r_k^*$ are average rewards that correspond to $r_k$. The difference between them is that $r_k^*$ is the optimal average reward in an MDP with the true dynamics $P$ and $\tilde{r}_k$ is the optimal average reward in an MDP with the optimistic dynamics $\tilde{P}_k$. Thus, the fact that the reward is known, in our case, does not change the fact that that the optimstic reward is a function of the dynamics uncertainty set $\mathcal{P}_k$.

To compute the optimstic policy and dynamics, UCRL2 uses the extended value iteration procedure of [63] to efficiently compute the following iterations:

$$u_0(s) = 0 \tag{13}$$

$$u_{i+1}(s) = \max_{a \in A} \left\{ r_k(s,a) + \max_{P \in \tilde{P}_k} \sum_{s' \in S} P(s'|s,a)u_i(s') \right\},$$

Using Theorem 7 from [35] we have that running extended value iteration to find the optimistic policy in the optimistic MDP for $t_k$ iterations guarantees that $\tilde{r}_k \geq r_k^* - 1/\sqrt{t_k}$. Thus, we have that:

$$\Delta_k \leq \sum_{s,a} v_k(s,a)(r_k^* - r_k(s,a)) + \sqrt{K} \leq \sum_{s,a} v_k(s,a)(\tilde{r}_k - r_k(s,a)) + 1/\sqrt{t_k} + \sqrt{K}$$

Using Eq. (13), we write the last iteration of the extended value iteration procedure as:

$$u_{i+1}(s) = r_k(s_k, \tilde{\pi}_k(s)) + \sum_{s' \in S} \tilde{P}_k(s'|s, (\tilde{\pi}_k(s)))u_i(s') \tag{14}$$

Theorem 7 from [35] guarantees that after running extended value iteration for $t_k$ we have that

$$\|u_{i+1}(s) - u_i(s) - \tilde{r}_k\| \leq 1/\sqrt{t_k}. \tag{15}$$

Plugging Eq. (14) in Eq. (15) we have that:

$$\|r_k(s_k, \tilde{\pi}_k(s)) - \tilde{r}_k + \sum_{s' \in S} \tilde{P}_k(s'|s, (\tilde{\pi}_k))u_i(s') - u_i(s)\| \leq 1/\sqrt{t_k}, \tag{16}$$

and therefore

$$\tilde{r}_k - r_k(s_k, a_k) = \tilde{r}_k - r_k(s_k, \tilde{\pi}_k(s)) \leq v_k(\tilde{P}_k - I)u_i + 1/\sqrt{t_k}.$$

In the next step in the proof, the vector $u_i$ is replaced with $w_k$, which is later upper bounded by the diameter of the MDP $D$. To conclude, we have that

$$\Delta_k \leq \sum_{s,a} v_k(s,a)(\tilde{r}_k - r_k(s,a)) + 1/\sqrt{t_k} + \sqrt{K} \leq v_k(\tilde{P}_k - I)w_k + 2/\sqrt{t_k} + \sqrt{K}.$$

From this point on, the proof follows by bounding the term $v_k(\tilde{P}_k - I)w_k$, which is only related to the dynamics, and combines all of the previous results into the final result, thus, it is possible to follow the original proof without any modification. Since the leading terms in the original proof come from uncertainty about the dynamics, we obtain the same bound as in the original paper.

## F   Experiments

Above, we presented a principled approach to using standard RL algorithms to solve convex MDPs. We also suggested that DRL agents can use this principle and solve convex MDPs by optimizing the reward from the cost player. We now demonstrate this by performing experiments with Impala [19], a distributed actor-critic DRL algorithm. Our main message is that in domains where Impala can solve RL problems (*eg*, problems without hard exploration), it can also solve convex MDPs.

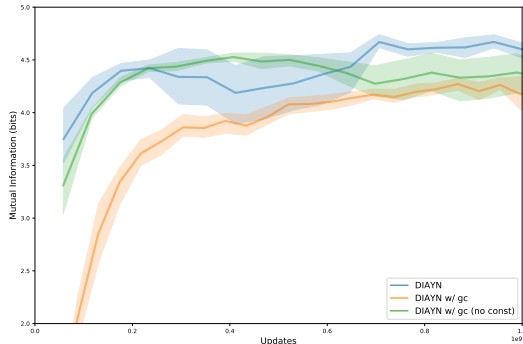

Figure 2: DIAYN, non uniform prior.

### F.1   DIAYN.

In our first experiment, we focus on the convex MDP formulation of DIAYN as we defined in Eq. (10). We compare the intrinsic reward that results from an FTL cost player in Eq. (11) and the original mutual-information based reward in DIAYN by performing ablative analysis on the gradient correction terms in Eq. (11). In both cases, we also include the standard action entropy regularizer. Since the two intrinsic rewards were shown to be equivalent under a uniform prior, we consider a fixed but non-uniform prior.[2] The environment is a simple $9 \times 9$ gridworld, where the agent can move along the four cardinal directions. We maximize undiscounted rewards over episodes of length 32. Given trajectories generated by the distributed actors, a central learner computes the gradients and updates the parameters for the policy, critic and the (variational) reverse predictor. Fig. 2 plots the average (per timestep) mutual information $I(z, s)$, between code $z$ and states $s \sim d_\pi^z$, which is equivalent to the objective in Eq. (10). Performance is averaged over 10 seeds, with the shaded area representing the standard error on the mean. Inspecting Fig. 2 we can see that DIAYN reaches around 4.5 bits. We can also see that using the full gradient correction term in Eq. (11) ("DIAYN w/ gc") degrades performance both in terms of convergence and final performance. On the other hand, removing the constant from the gradient correction ("DIAYN w/ gc (no const)"), which does not affect the optimal policy, recovers the performance of DIAYN.

### F.2   Entropy constrained RL.

Here we focus on an MDP with a convex constraint, where the goal is to maximize the extrinsic reward provided by the environment with the constraint that the entropy of the state-action occupancy measure must be bounded below. In other words, the agent must solve $\max_{d_\pi \in \mathcal{K}} \sum_{s,a} r(s, a) d_\pi(s, a)$ subject to $H(d_\pi) \geq C$, where $H$ denotes entropy and $C > 0$ is a constant. The policy that maximizes the entropy over the MDP acts to visit each state as close to uniformly often as is feasible. So, a solution to this convex MDP is a policy that, loosely speaking, maximizes the extrinsic reward under the constraint that it explores the state space sufficiently. The presence of the constraint means that this is not a standard RL problem in the form of Eq. (1). However, the agent can solve this problem using the techniques developed in this paper, in particular those discussed in Section 5.

We evaluated the approach on the bsuite environment 'Deep Sea', which is a hard exploration problem where the agent must take the exact right sequence of actions to discover the sole positive reward in the environment; more details can be found in [53]. In this domain, the features are one-hot state

---

[2]$p(z)$ is a Categorical distribution over $n = 2^5$ outcomes, with $p(z = i) = u_i / \sum_{j=1}^n u_j$, $u_i \sim U(0, 1)$.

features, and we estimate $d_\pi$ by counting the state visitations. For these experiments we chose $C$ to be half the maximum possible entropy for the environment, which we can compute at the start of the experiment and hold fixed thereafter. We equipped the agent with the (non-stationary) Impala algorithm, and the cost-player used FTL. We present the results in Figure 3 where we compare the basic Impala agent, the entropy-constrained Impala agent and bootstrapped DQN [52]. As made clear in [50] algorithms that do not properly account for uncertainty cannot in general solve hard exploration problems. This explains why vanilla Impala, considered a strong baseline, has such poor performance on this problem. Bootstrapped DQN accounts for uncertainty via an ensemble, and consequently has good performance. Surprisingly, the entropy regularized Impala agent performs approximately as well as bootstrapped DQN, despite not handling uncertainty. This suggests that the entropy constrained approach, solved using Algorithm 1, can be a reasonably good heuristic in hard exploration problems.

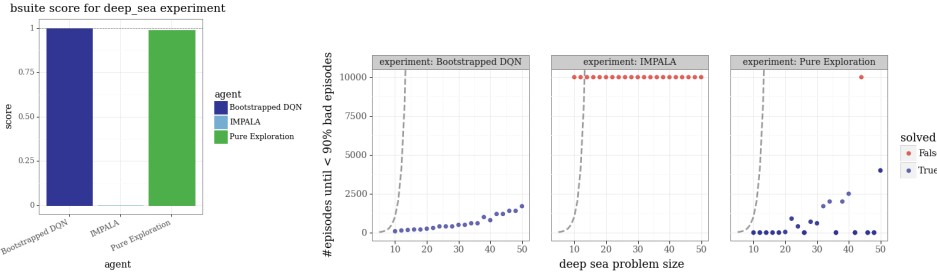

Figure 3: Entropy constrained RL