# OpenReview forum: "Reward is enough for convex MDPs"
_NeurIPS.cc/2021/Conference — NeurIPS 2021 Spotlight_

### Official Review · Reviewer_3z88 · 2021-07-06

**Rating:** 7
**Confidence:** 3

**Summary:**

This paper addresses an extended reinforcement learning setting with general utilities, where the agent's objective can be any convex function of the state-action distribution. Especially, it recasts the problem as a min-max game between a policy player and a cost player, and it describes a principled methodology to solve the game. Finally, it provides ways to instantiate the methodology with known algorithms, and a brief experimental evaluation in simple domains.

**Limitations And Societal Impact:**

The work is mainly theoretical, and it does not need to address potential societal impacts. The limitations are biefly mentioned in the checklist and through the text.

**Main Review:**

*POST-REBUTTAL*

The score has been updated for the reasons explained in the comment below.

------


This paper studies a very relevant problem, which practically unifies many RL objectives of interest under a unique setting, and it provides a neat formulation of the framework as a two-player min-max game. However, the major takes of this paper are not particularly surprising w.r.t. prior works, and some choices in the presentation of the practical algorithms are quite confusing. Thus, I am currently evaluating this work as borderline, though a convincing authors' response and slight changes to the main text might better clarify the value of the paper.
I provide below some detailed comments.

TECNICHAL NOVELTY

The Convex MDP framework has been previously formulated in (J. Zhang et al., 2020). The meta-algorithm and its analysis are taken from (Abernethy and Wang, 2017). The FTL and best response combination is practically equivalent to the algorithm in (Hazan et al., 2019). Even the min-max game interpretation, which is novel to my knowledge, is somehow suggested by the work in (J. Zhang et al., 2020), where the gradient computation requires to solve a max-min problem between the policy update and the reward. I believe that this work successfully meshes these ingredients in a more coherent presentation (Sect.1 to Sect.3), but I have some doubts on the actual novelty of the contributions.
Can the authors clearly specify the technical novelty their work is contributing w.r.t. prior works?

PRESENTATION AND CONTRIBUTIONS

I found the paper quite neat and clear from Sect.1 to Sect.3, whereas it becomes somewhat confusing thereafter. Instead of the various alternatives provided by Sect.4, I would have rather focused on a single algorithm to tackle convex RL in an online setting, then providing either a complete theoretical analysis or a more thorough experimental evaluation in the next sections. Especially, some of the questions that could be targeted are:
- There exists any separation between the learning complexity of the Convex MDP setting w.r.t. the standard MDP setting?
- Can we deploy a single algorithm that is able to address the variety of objectives reported in Table 1 in a practical way?

GENERAL METHODOLOGY

It is not completely clear to me why different objectives should require different implementations of the policy player and the cost player, rather than just considering the Fenchel dual of the respective objective function. The authors rightly noted that the algorithm from (J. Zhang et al., 2020) is committing to a specific methodology (policy gradient), but a key feature of their approach is that it can be employed with any convex objective.
Can the authors clarify why they are not providing an objective-agnostic implementation of Algorithm 1?

**Time Spent Reviewing:**

6

---

> ### Author Response · Authors · 2021-08-09
> **Author response.**
>
> We thank the reviewer for finding our paper to be relevant and to provide a neat formulation to an important problem. We hope that our rebuttal will help to clarify the novelty of our work and convince the reviewer that our work warrants their acceptance. Please also check our main author response which includes our response to your last question.
>
> As the reviewer suggested, the convex MDP formulation was already proposed in (Hazan et al. 2019) and was extended to include constraints in (J. Zhang et al., 2020). In fact, many of the algorithms that were developed to solve the Aprenetienchip Learning problem, can solve any other convex RL problem (e.g. Abbeel and Ng 2004 or Syed and Schapire 2007), since they are instances of Algorithm 1. Our contributions on this front are the following: (a) we extended the constraint MDP formulation to include convex constraints (while previous work focused on linear constraints). (b) we analyzed the non convex case and showed that diversity maximizing algorithms can be viewed as solving a concave RL problem. While the theoretical guarantees in this case are weaker, these methods perform quite well in practice and are actively being explored. Thus we hope that putting them under the same framework is a worthwhile contribution.
>
> More importantly, while previous work already identified the convex RL problem, each paper focused on a single algorithm as a solution and the connection between the different algorithms was loose. Our meta algorithm (adapted from Abernethy and Wang, 2017) unifies the different algorithms in the convex RL landscape into a single framework.
>
> On the technical front, the reviewer is correct that a min-max formulation via Fenchel duality was also suggested by (J. Zhang et al., 2020) as we discussed in lines 136-142. However, in this paper Fenchel duality was used in a different manner. Concretely, the authors first defined the variational policy gradient and then showed that in order to compute this gradient, it is needed to solve a saddle point problem via Fenchel duality. The caveat in this approach is that the algorithm requires solving a saddle point problem in each iteration. In that sense it is similar to FW algorithms that have to find the best response in each iteration, since both sub problems are challenging.
>
> By contrast, in our paper we define the convex RL problem as solving a single saddle point problem. On the whole this is not particularly novel since we borrowed the analysis from (Abernethy and Wang, 2017) and applied it to the convex RL problem, but its implications are of great importance to the RL community.
>
> In particular, we were able to show that one can use any RL algorithm as a policy player. For RL algorithms like UCRL2 and MDPO we proved a $\sqrt{T}$ regret bound, and showed that it leads to an O($1/\epsilon^2$) sample complexity bound for the convex RL problem via a standard online to batch conversion [Cesa-Bianchi et al. 2004], i.e., for convex RL problems our approach requires O($1/\epsilon^2$) interactions with the environment to read $\epsilon$ accuracy.
>
> This is an important novelty of our work for the following reasons. Previous algorithms with theoretical results for convex MDPs focused on best response policy players and gave the false impression that the convex RL problem is harder than the standard RL problem (one has to solve a standard RL problem in each iteration). These results typically ignored the amount of interactions with the environment that are required to find the best response. When these interactions were included in the sample complexity analysis, the previously best known result suggested that it is enough to have O($1/\epsilon^3$) interactions with the environment to read epsilon accuracy [Hazan et al. 2019]. In addition, using RL algorithms as policy players is conceptually very simple, and much closer to what is being done in practical applications.
>
> The sample complexity of the standard RL problem is also known to be O($1/\epsilon^2$), see, for example, [Dann and Brunskill, 2016], for matching upper and lower bounds. Thus, not only that we improved the best known sample complexity for the convex RL problem, our results also suggest that convex RL is as hard as the standard RL problem in terms of $\epsilon$ (which answers the reviewers question). We will make sure to clarify that contribution in the paper. We note that this sample complexity bound is perhaps not tight in the other constants (e.g., $\delta$ and |S|), so it remains an open question if the two problems are equivalently hard in all the constants.
>
> Sample Complexity of Episodic Fixed-Horizon Reinforcement Learning, Christoph Dann and Emma Brunskill, NeurIPS 2016.
>
> N. Cesa-Bianchi, A. Conconi, and C. Gentile. On the generalization ability of on-line learning algorithms. Information Theory, IEEE Transactions on, 50(9):2050–2057, Sept. 2004.
>
>
>
>
>
>
> Copied from the main author response. The following questions repeated themselves by reviewers 3z88 & WgJuP. It is an important question and our answer to it can also be useful for other reviewers. Thus, we address it in our global response below and add a discussion around it in the main paper. Question: Which algorithm would you recommend to solve a general convex RL problem? Should one use different algorithms in different problems?
>
> We would like to start by stating that in Section 5 we are not suggesting to use a different combination of the two players in different convex RL problems, but rather, that existing algorithms in the convex RL literature, can be explained as instances of Algorithm 1 by choosing  different combinations of the two players. That said, it is interesting to observe in hindsight that different algorithms were preferred in different problems over the years.
>
> We would like to emphasize that any of the algorithms proposed in the paper can be used to solve any convex RL problem. That said, there are different considerations, from the theoretical and practical points of view that one should consider when choosing an algorithm for his problem.
>
> Sample complexity: On this front, we have shown that different policy players result in different worst case sample complexity guarantees as a function of the desired accuracy $\epsilon$. In particular, we showed that using an RL algorithm with non stationary reward is more sample efficient (requires O($1/\epsilon^2$) samples) than an approximate best response that requires O($1/\epsilon^3$) samples). See the discussion above for more detail.
>
> Iteration complexity: Different algorithms enjoy different iteration complexity and will be preferred in different situations. For example, in the AL literature, many algorithms use the best response as a policy player (solve an MDP in each iteration) and are compared on the basis of the number of iterations that they require in order to find an $\epsilon-$optimal solution (as a function of d, the feature dimension and $\epsilon$). For example, Abbeel and Ng (2004) have iteration complexity of O($d\log(1/\epsilon)$), vs Syed and Schapire (2007) which enjoys O($\log(d)/\epsilon^2$). Thus, a different algorithm will be preferred for different values of $d$ and $\epsilon$.
> Practical recommendation: In order to solve a practical convex RL problem in a given domain, our advice would be to start from an RL algorithm that is known to be high performing in the standard RL problem in this domain. From the theoretical point of view, this should be MDPO or UCRL2, for which we have proved a regret bound. From the practical point of view, we would recommend using a high performing DRL algorithm, which may be specific to the domain.
>
> Our advice would then be to use this DRL algorithm as an RL player with a non stationary reward, and not as an (approximate) best response. This design choice is inspired by its simplicity, from our sample complexity analysis and from the fact that it has been adapted by practitioners in various problems (e.g. GAIL, DIAYN).
>
> For the cost player, we would recommend FTL, i.e., to compute the gradient of the objective, for its simplicity. However there remains a choice of how best to estimate the state occupancy, which will vary based on the precise nature of the convex function f.

---

> > ### Comment · Reviewer_3z88 · 2021-08-24
> > **Post-rebuttal comment**
> >
> > I would like to thank the authors for their thorough replies.
> >
> > I was previously evaluating this work as borderline due to its limited novelty. However, I was overlooking the importance of the sample complexity result, which I think is a core contribution of this paper and might open the door to further analysis on the separation between the convex and linear MDP problems. Moreover, I perhaps underestimated the value of unifying some of the previous results under an accessible framework, which might have a positive impact on future research as well.
> >
> > For these considerations, I am raising my score and I am now recommending accepting the paper.
> >
> > I provide below some suggestions the authors may consider in an updated version of the paper:
> > - As the authors noted in their response, the sample complexity result could be better highlighted in the paper, and I would consider to mention it in the abstract;
> > - The fact that Section 5 is basically a survey of existing methodologies under the unified framework could be clarified, as the authors did in their response;
> > - The figure in the experimental section is not looking particularly good, it is not easy to read the labels and the legend, and the caption is missing altogether;
> > - I point a couple of references the authors might find relevant to their paper, which are (Geist et al., Concave utility reinforcement learning, 2021) and (Husain et al., Regularized policies are reward robust, 2021). The first one is concurrent to this work and might have overlapping contributions.
> >
> > https://arxiv.org/abs/2106.03787
> >
> > http://proceedings.mlr.press/v130/husain21a/husain21a.pdf

---

### Official Review · Reviewer_WgJu · 2021-07-14

**Rating:** 7
**Confidence:** 2

**Summary:**

The paper discusses the problem of minimizing a convex function of the stationary distribution of a system with Markov dynamics, subject to the (convex) Bellman flow constraint and optionally subject to other convex constraints. The paper provides a family of algorithms designed to tackle this problem. It also proves results about sample complexity. The proposed framework is a significant innovation since it unifies several apprenticeship learning and imitation learning algorithms, which have been proved to be useful.

**Limitations And Societal Impact:**

The authors claim that there is no need to discuss societal implications since this is a theoretical paper. Because of the breath of applications that the paper covers, the discussion would be so inspecific that I tend to agree.

**Main Review:**

The submission is original and high-quality. It is clearly written despite the breadth of material it covers. The contribution is significant and a good match for NeurIPS.

I have the following questions for the authors:
1. As a practitioner, I am interested in concrete algorithmic recommendations. The paper lists several choices of algorithms that adjust $\lambda$ and several that adjust $\pi$, but at the end of the day, given a concrete problem of the form (2), how do I decide which of these to use? A discussion would be really nice.
2. In lines 162,163 you say that supporting CMDPs (and other problems with constraints) requires a minor adjustment to the algorithm, but you don't specifically define what adjustment. Can you clarify how Algorithm 1 changes?

I would also suggest removing the experimental section from the main paper and putting it in the appendix - the strength of your submission lies in the generality of the theoretical contribution.

Minor points:
1. The figures on page 9 are completely illegible on a printed copy. Please enlarge and move to the appendix.
2. The checklist should be part of the main paper.
3. The notations $f^\star$ for the optimum value and $f^*$ for the Fenchel conjugate are so similar as to be confusing. I would rename the optimum value.
4. I don't think Figure 1 is very useful. To put it bluntly, people who can understand the rest of the paper are unlikely to need it.
5. Line 603 in the appendix: "[31, equation 9 1]" should be "[31, equation 9]".

**Time Spent Reviewing:**

5

---

> ### Author Response · Authors · 2021-08-09
> **Author response.**
>
> We thank the reviewer for their kind words and positive feedback. The reviewer raised two important points. Please see our response in the main author response for your first question, which we also copied below for your convenience.
>
> Regarding the second question: Note that the Lagrangian in line 159 is convex in $d_\pi$ and concave in $\nu$,$\mu$ and $\zeta$. Thus, finding this saddle point can be done with the same techniques which we developed in the paper. Concretely, this will require extending the cost player to include the three variables $\nu,$ $\mu$ & $\zeta$ and applying any OCO algorithm as a cost player for these three variables.
>
> However, further considerations need to be taken care of when one wants to use the additional structure in this Lagrangian (arising from Fenchel duality) as was used in the unconstrained case for the FTL player. The $\nu$ variable is equivalent to $\lambda$ and independent from the from $\zeta$ and $\mu$ variables, thus, the FTL player for $\nu$ will be the gradient of f as before. For $\zeta$ and $\mu$ the situation is slightly more complicated as they appear jointly in $g^*$ (the Fenchel conjugate of the constraint) via the fraction $\frac{\zeta}{\mu}$. In this case, the FTL player for $\mu$ and $\zeta$ it will be the gradient of the conjugate of the function $h(\zeta, \mu) = \mu g^* (\zeta / \mu)$. We will add these details to the final version of the paper.
>
> As the reviewer suggested, we will move the experiments to the supplementary material, enlarge the figures and include the checklist in the main paper. We will also change the notation for the optimum value of f to OPT.
>
> Copied from the main author response:
> The following questions repeated themselves by reviewers 3z88 & WgJuP. It is an important question and our answer to it can also be useful for other reviewers. Thus, we address it in our global response below and add a discussion around it in the main paper. Question: Which algorithm would you recommend to solve a general convex RL problem? Should one use different algorithms in different problems?
>
> We would like to start by stating that in Section 5 we are not suggesting to use a different combination of the two players in different convex RL problems, but rather, that existing algorithms in the convex RL literature, can be explained as instances of Algorithm 1 by choosing  different combinations of the two players. That said, it is interesting to observe in hindsight that different algorithms were preferred in different problems over the years.
>
> We would like to emphasize that any of the algorithms proposed in the paper can be used to solve any convex RL problem. That said, there are different considerations, from the theoretical and practical points of view that one should consider when choosing an algorithm for his problem.
>
> Sample complexity: On this front, we have shown that different policy players result in different worst case sample complexity guarantees as a function of the desired accuracy $\epsilon$. In particular, we showed that using an RL algorithm with non stationary reward is more sample efficient (requires O($1/\epsilon^2$) samples) than an approximate best response that requires O($1/\epsilon^3$) samples). See the discussion above for more detail.
>
> Iteration complexity: Different algorithms enjoy different iteration complexity and will be preferred in different situations. For example, in the AL literature, many algorithms use the best response as a policy player (solve an MDP in each iteration) and are compared on the basis of the number of iterations that they require in order to find an $\epsilon-$optimal solution (as a function of d, the feature dimension and $\epsilon$). For example, Abbeel and Ng (2004) have iteration complexity of O($d\log(1/\epsilon)$), vs Syed and Schapire (2007) which enjoys O($\log(d)/\epsilon^2$). Thus, a different algorithm will be preferred for different values of $d$ and $\epsilon$.
> Practical recommendation: In order to solve a practical convex RL problem in a given domain, our advice would be to start from an RL algorithm that is known to be high performing in the standard RL problem in this domain. From the theoretical point of view, this should be MDPO or UCRL2, for which we have proved a regret bound. From the practical point of view, we would recommend using a high performing DRL algorithm, which may be specific to the domain.
>
> Our advice would then be to use this DRL algorithm as an RL player with a non stationary reward, and not as an (approximate) best response. This design choice is inspired by its simplicity, from our sample complexity analysis and from the fact that it has been adapted by practitioners in various problems (e.g. GAIL, DIAYN).
>
> For the cost player, we would recommend FTL, i.e., to compute the gradient of the objective, for its simplicity. However there remains a choice of how best to estimate the state occupancy, which will vary based on the precise nature of the convex function f.

---

### Official Review · Reviewer_wKfh · 2021-07-19

**Rating:** 7
**Confidence:** 5

**Summary:**

The paper studies the convex RL problem, which is defined as finding policy $\pi$ minimizing $f(d_\pi)$ where $f$ is a convex function and $d_\pi$ is the occupancy measure of policy $\pi$. Using Fenchel duality, this formulation can be written as a min-max and be solved using the well-known framework of repeated game playing first introduced by Freund and Schapire (1999). The authors show that their framework unifies some well-known approaches in RL by appropriately choosing the no-regret algorithm for the policy and cost player.

**Limitations And Societal Impact:**

The limitations and societal impact have been adequately addressed.

**Main Review:**

**Originality and Quality**

The proof techniques and theoretical components of the paper already exist in the prior work. However, the paper's main contribution is providing a unifying framework that captures previous approaches through a single lens of convex RLs.  Although the materials already exist in the prior work, they are scattered, and the authors did a great job of bringing all of them under a single framework.

I believe the paper's main contribution is the framework itself and its connections that have been well discussed. Therefore, I would suggest moving the experiments to the appendix as they are not adding to the paper's contribution.

The proofs and claims are sound and well-supported.

**Clarity**

I have much enjoyed reading the paper as it's very well-written and well-organized.

**Significance**

The paper will be of interest to many researchers in the RL community.

**Time Spent Reviewing:**

2

---

> ### Author Response · Authors · 2021-08-09
> **Authors response.**
>
> We thank the reviewer for their kind words. Please see our clarification of technical novelty in the main author's response. For your convenience, we also briefly summarised it below and copied our answer from the main author response below it. We are also happy to move the experiments to the appendix as we mentioned in the main response. Please let us know if there are further questions.
>
> Technical novelty. We agree with the reviewer that many of the materials are scattered across the literature, and providing a unifying framework was indeed part of our original motivation. We do believe however that our paper offers several additional contributions. We extended the scope of policy players from best response [Hazan et al] to incorporate non-stationary RL algorithms (variants of MDPO and UCRL2) and extended the analysis of approximate best response to a general cost player (Hazan et al pioneered this direction, but only for the FTL player via approximate FW analysis). In the process, we also showed that using RL algorithms as policy players leads to a sample complexity result of $1/\epsilon^2$, which not only improves on the previously known result of $1/\epsilon^3$ [Hazan et al. 2019] but also matches the best known result for the standard RL problem. A surprising and powerful result in our opinion. We note that this sample complexity bound is perhaps not tight in the other constants (e.g., $\delta$ and |S|), so it remains an open question if the two problems are equivalently hard in all the constants.
>
> ##### Copied from the main author response.
> We want to emphasise a result in our paper that we feel we neglected to highlight in the original submission. We proved that solving the convex RL problem to accuracy epsilon requires O($1/\epsilon^2$) samples. The result is based on Equation 5 in our paper, lemma 3, lemma 4 and the discussion in lines 242-247. The standard RL problem (which is a special case of the convex RL problem) has matching O($1/\epsilon^2$) upper and lower bounds (we added a formal statement of this to the draft based on previous work). So we now have matching upper and lower bounds for the convex RL problem of O($1/\epsilon^2$), the first paper to produce these to the best of our knowledge. Previously, the best result for the convex RL problem required O($1/\epsilon^3$) samples [Hazan et al. 2019], so we have improved up that and shown that the convex RL problem is not harder than the standard RL, but is in fact the same hardness (in terms of epsilon at least). We note that this sample complexity bound is perhaps not tight in the other constants (e.g., $\delta$ and $|S|$), so it remains an open question if the two problems are equivalently hard in all the constants. We have rewritten the abstract and introduction to highlight this result, and made it a formal theorem in the paper. We also referred to this result in our responses to the individual reviewers, and in particular, to reviewer 3z88 where we discuss it in more detail.

---

> > ### Comment · Reviewer_wKfh · 2021-08-26
> > **Post-rebuttal**
> >
> > I want to thank the authors for providing a detailed response to me and the other reviewers. After reading the evaluations and the author response, I still vote for acceptance and keep my score.

---

### Official Review · Reviewer_BxxS · 2021-07-19

**Rating:** 6
**Confidence:** 4

**Summary:**

The authors propose to consider a convex optimization problem over the polytope formed by the state-visitation frequency of an MDP. They constructed algorithms based on existing tools of sub-gradient descents, such as Frank-Wolfe, FTL and OMD, in conjunction with an oracle that (approximate)-solves an MDP with scalar rewards. Altogether, they provide a meta-algorithm that translates a scalar reward MDP oracle to an algorithm for the convex MDP. The authors also highlight several applications, such as the Apprenticeship Learning problem and the constrained MDP problem, of their formulation.

**Main Review:**

While the authors propose an interesting problem, I have a major concern over the novelty of the paper as compared to the following SODA 2015 paper by Agrawal and Devanur 2015:

https://arxiv.org/abs/1410.7596

The authors' convex optimization can be phrased in their problem setting (in their Definition 1), by putting $A_t = \mathcal{K}$ for all $t$. (And also rephrasing convex minimization as concave maximization). Algorithm 5.1 in (Agrawal and Devanur 2015) solves convex optimization in Definition 1. The algorithm only requires an oracle that outputs solution to a linear optimization (the definition of $v^\dagger_t$ in Algorithm 5.1)  over the feasible region $A_t$. The framework in the (Agrawal and Devanur 2015) is quite similar to what is proposed in the paper in the sense that both allow a wide suit of (sub)-gradient descent algorithms, as highlighted in their Section 3. In this regard, the algorithm design and analysis of Algorithm 1 appear to be fundamentally based on existing works.

A difference between the two papers would be that, while Agrawal and Devanur 2015 assume an exact solver in their generation of $v^\dagger_t$, while the submission allows an approximate solver. Another is that, the submission proposes the use of non-stationary RL algorithms, and they also highlighted how certain objective (for example, the objective based on mutual information in Section 5.2) can be cast under their framework. Nevertheless, in my opinion Algorithm 1 is the main technical part of the paper, and the two differences highlighted previously are secondary in comparison (this is not to say that they are unimportant, but I saw Algorithm 1 in the paper as the main tool that solves the proposed problem).

In addition, there is also a minor concern on the presentation of Algorithm 1. The authors display the Algorithm 1, which generates a sequence of occupancy measures $d^k_{\pi}$ for $k = 1, \ldots, K$. For any occupancy measure $d$, it is asserted in Line 73 that they recover a policy $\pi$ that generates $d$ by setting $\pi(s, a) = \frac{d(s, a)}{\sum_{a\in A}d(s, a)}$. But what if $\sum_{a\in A}d(s, a) = 0$ for a state $s$? The authors should address this technicality in Line 73. Nevertheless, I do not think it is a technical issue for their Algorithm 1, in the sense that the authors first outputs a policy $\pi_k$ that solves the maximizing objective when the scalarization is $-\lambda^k$, only then they report the occupancy measure of $\pi_k$ as $d^k_{\pi}$.

**Time Spent Reviewing:**

6

---

> ### Author Response · Authors · 2021-08-09
> **Authors response**
>
> We would like to thank the reviewer for bringing to our attention the SODA 2015 paper by Agrawal and Devanur 2015. This is a very interesting paper and is highly relevant to our work, but unfortunately we missed it during our literature review. As the reviewer mentioned, the paper proposes to solve Online Convex Optimization (OCO) by applying a primal-dual algorithm to the Lagrangian which results from Fenchel duality. In that sense, the mentioned paper addresses a similar problem to the Abernethy & Wang 2017 paper, but preceded it. We will make sure to clarify that point in our paper.
>
> In their review, the reviewer implied that Algorithm 1 is our main technical novelty. We kindly disagree with this assessment as we clearly emphasized in the paper that Algorithm 1 is borrowed from Abernethy & Wang 2017 but applied to the convex RL problem and is therefore not the main technical novelty of our paper. Our novelty, w.r.t. Algorithm1, is adapting it to the convex RL context and providing a unifying framework encompassing many existing algorithms in the literature. In addition, we would like to emphasize that the convex RL problem is not a standard OCO problem since not all standard OCO algorithms will work as a policy player. The reason for that is that while the convex RL problem is convex in the state occupancy $d_\pi$, it is not convex in the policy $\pi$ itself.
>
> The exception is the best response, which is an OCO algorithm, but is challenging to use in practice for convex RL problems, as one has to solve a full RL problem in each iteration. The focus on best response algorithms in the literature also gives the false impression that the convex RL problem is more complicated than the standard RL problem. For that reason, and as the reviewer mentioned, our paper departs from the Agrawal and Devanur 2015 paper, by considering a wider variety of algorithms as policy players (Approximate Best Response, Non-Stationary RL).
>
> More specifically, we analyzed RL algorithms as policy players, which are not OCO algorithms, and adapted them to the non stationary reward setup. We believe that this is the main technical novelty of our work, compared to previous work which focused on best response. In addition, we showed that using RL algorithms as policy players leads to a sample complexity result of $1/\epsilon^2$. Not only does it improve on the previously best known sample complexity result for the convex RL problem ($1/\epsilon^3$ for approximate best response, Hazan et al. 2019), it also matches the best known result for the standard RL problem. We note that this sample complexity bound is perhaps not tight in the other constants (e.g., $\delta$ and $|S|$), so it remains an open question if the two problems are equivalently hard in all the constants.
>
> Regarding $d_\pi$, the reviewer raises an important issue which arises when there is a transient state in the MDP and $\sum_a d_\pi(s,a)=0$. This situation can be handled by defining the policy to be uniform in such states, i.e.: $\pi(s,a)$ = 1/num_actions when  $\sum_a d_\pi(s,a)=0$. This is an arbitrary choice and it is not affecting the algorithm (since the state is transient, it is visited with probability 0). We would clarify this point in the paper. In addition, as the reviewer suggested, in line 73 we only mentioned that it is possible to recover the policy from its state occupancy, and indeed Algorithm 1 is not using this property.

---

> > ### Comment · Reviewer_BxxS · 2021-08-16
> > **Thanks for the clarification on the main contributions**
> >
> > I thank the authors for the clarification regarding the contribution on Algorithm 1, and I also acknowledge that the authors have already attributed the use of OCO to Abernethy and Wang 2017. I believe that the main novelty here is that the authors bring together a wide variety of OCO tools and RL algorithms to solve the convex RL problem, and also provide a list of examples. In addition, the authors also propose ways to save samples by a careful consideration on solving a sequence of related non-stationary MDPs. I also agree that the authors deviate from the default choice of best response for the policy player, and the authors propose some careful approaches to refine the sample complexity. Correspondingly, I increase the score. The authors' effort in bringing a variety of OCO tools and refining the use of the approximate response is a great effort, and the paper is well executed and clearly written.

---

### Author Response · Authors · 2021-08-09
**Author response**

We would like to thank the reviewers for spending time reading our paper and providing us feedback. We enjoyed reading the reviews and we believe that they will help us improve the quality of our paper. We hope that our rebuttal addresses all of your concerns, but if we missed anything, please follow up with more questions.

We want to emphasise a result in our paper that we feel we neglected to highlight in the original submission. We proved that solving the convex RL problem to accuracy epsilon requires O($1/\epsilon^2$) samples. The result is based on Equation 5 in our paper, lemma 3, lemma 4 and the discussion in lines 242-247. The standard RL problem (which is a special case of the convex RL problem) has matching O($1/\epsilon^2$) upper and lower bounds (we added a formal statement of this to the draft based on previous work). So we now have matching upper and lower bounds for the convex RL problem of O($1/\epsilon^2$), the first paper to produce these to the best of our knowledge. Previously, the best result for the convex RL problem required O($1/\epsilon^3$) samples [Hazan et al. 2019], so we have improved up that and shown that the convex RL problem is not harder than the standard RL, but is in fact the same hardness (in terms of epsilon at least). We note that this sample complexity bound is perhaps not tight in the other constants (e.g., $\delta$ and $|S|$), so it remains an open question if the two problems are equivalently hard in all the constants. We have rewritten the abstract and introduction to highlight this result, and made it a formal theorem in the paper. We also referred to this result in our responses to the individual reviewers, and in particular, to reviewer 3z88 where we discuss it in more detail.

Minor: Reviewers wKfh and WgJu suggested to move the experiments to the supplementary material to allow more space for other content. Since an extra page is allowed in the final version of the paper we believe that we will be able to include both new content and the experiments in the final version. Otherwise, we will move the experiments as the reviewers suggested.

The following questions repeated themselves by reviewers 3z88 & WgJuP. It is an important question and our answer to it can also be useful for other reviewers. Thus, we address it in our global response below and add a discussion around it in the main paper. Question: Which algorithm would you recommend to solve a general convex RL problem? Should one use different algorithms in different problems?

We would like to start by stating that in Section 5 we are not suggesting to use a different combination of the two players in different convex RL problems, but rather, that existing algorithms in the convex RL literature, can be explained as instances of Algorithm 1 by choosing  different combinations of the two players. That said, it is interesting to observe in hindsight that different algorithms were preferred in different problems over the years.

We would like to emphasize that any of the algorithms proposed in the paper can be used to solve any convex RL problem. That said, there are different considerations, from the theoretical and practical points of view that one should consider when choosing an algorithm for his problem.

Sample complexity: On this front, we have shown that different policy players result in different worst case sample complexity guarantees as a function of the desired accuracy $\epsilon$. In particular, we showed that using an RL algorithm with non stationary reward is more sample efficient (requires O($1/\epsilon^2$) samples) than an approximate best response that requires O($1/\epsilon^3$) samples). See the discussion above for more detail.

Iteration complexity: Different algorithms enjoy different iteration complexity and will be preferred in different situations. For example, in the AL literature, many algorithms use the best response as a policy player (solve an MDP in each iteration) and are compared on the basis of the number of iterations that they require in order to find an $\epsilon-$optimal solution (as a function of d, the feature dimension and $\epsilon$). For example, Abbeel and Ng (2004) have iteration complexity of O($d\log(1/\epsilon)$), vs Syed and Schapire (2007) which enjoys O($\log(d)/\epsilon^2$). Thus, a different algorithm will be preferred for different values of $d$ and $\epsilon$.
Practical recommendation: In order to solve a practical convex RL problem in a given domain, our advice would be to start from an RL algorithm that is known to be high performing in the standard RL problem in this domain. From the theoretical point of view, this should be MDPO or UCRL2, for which we have proved a regret bound. From the practical point of view, we would recommend using a high performing DRL algorithm, which may be specific to the domain.

Our advice would then be to use this DRL algorithm as an RL player with a non stationary reward, and not as an (approximate) best response. This design choice is inspired by its simplicity, from our sample complexity analysis and from the fact that it has been adapted by practitioners in various problems (e.g. GAIL, DIAYN).

For the cost player, we would recommend FTL, i.e., to compute the gradient of the objective, for its simplicity. However there remains a choice of how best to estimate the state occupancy, which will vary based on the precise nature of the convex function f.

---

> ### Author Response · Authors · 2021-08-26
> **Post rebuttal**
>
> We would like to thank the reviewers for their thorough feedback, for engaging in the rebuttal, and for reconsidering their original evaluation of the paper. This is much appreciated and we believe that the paper gained a lot from their feedback. We will make sure to reflect all of the comments in the final version of the paper as we promised.
> -- the authors

---

### Decision · Program_Chairs · 2021-09-27

**Decision:**

Accept (Spotlight)

**Comment:**

This paper is very clearly written, brings together a large literature and using that unification to provide concrete conclusions about a few open problems about convex MDPs. The reviewers all agreed that the unification provided in this work is both very well done and useful to the community. It was also identified that the paper establishes an important sample complexity result: a rate of O(epsilon^{-2}) for convex MDPs, and that this contribution should be better highlighted. The author response helped allay reviewer concerns. More than one reviewer suggested moving the experiments to the appendix, and you could consider this to spend more time on the theoretical contributions. The reviewers each also gave useful suggestions for the work, and I highly encourage the authors to consider incorporating them.